# LIM Kinases, LIMK1 and LIMK2, Are Crucial Node Actors of the Cell Fate: Molecular to Pathological Features

**DOI:** 10.3390/cells12050805

**Published:** 2023-03-04

**Authors:** Elodie Villalonga, Christine Mosrin, Thierry Normand, Caroline Girardin, Amandine Serrano, Bojan Žunar, Michel Doudeau, Fabienne Godin, Hélène Bénédetti, Béatrice Vallée

**Affiliations:** 1Centre de Biophysique Moléculaire; UPR4301, CNRS, University of Orleans and INSERM, CEDEX 2, 45071 Orleans, France; 2Laboratory for Biochemistry, Department of Chemistry and Biochemistry, Faculty of Food Technology and Biotechnology, University of Zagreb, 10000 Zagreb, Croatia

**Keywords:** LIMK, actin dynamics, cytoskeleton remodelling, signalling pathways

## Abstract

LIM kinase 1 (LIMK1) and LIM kinase 2 (LIMK2) are serine/threonine and tyrosine kinases and the only two members of the LIM kinase family. They play a crucial role in the regulation of cytoskeleton dynamics by controlling actin filaments and microtubule turnover, especially through the phosphorylation of cofilin, an actin depolymerising factor. Thus, they are involved in many biological processes, such as cell cycle, cell migration, and neuronal differentiation. Consequently, they are also part of numerous pathological mechanisms, especially in cancer, where their involvement has been reported for a few years and has led to the development of a wide range of inhibitors. LIMK1 and LIMK2 are known to be part of the Rho family GTPase signal transduction pathways, but many more partners have been discovered over the decades, and both LIMKs are suspected to be part of an extended and various range of regulation pathways. In this review, we propose to consider the different molecular mechanisms involving LIM kinases and their associated signalling pathways, and to offer a better understanding of their variety of actions within the physiology and physiopathology of the cell.

## 1. Introduction

In 1994, LIM kinase 1 (LIMK1) was discovered simultaneously by the teams of Mizuno [1] and Bernard [2]. It was then described as the first kinase protein seen to contain LIM domains. The LIM kinase family was extended a year later with the discovery of LIM kinase 2 (LIMK2), which has a shared sequence of nearly 51% with LIMK1. Even though they are closely related, LIM kinases display cell-type-specific expression and different subcellular localisation [3]. LIM kinase expression patterns were first established in 2006 by Sumi and by Acevedo [3,4]: LIMK1 is particularly expressed in the brain, heart, skeleton muscles, kidneys, and lungs [5,6], while LIMK2 is more widely expressed in adult and embryonic tissues [4]. 

Canonically, LIM kinases act as downstream effectors of the members of the Rho GTPase family, including Rho, Rac and Cdc42, which modulate LIMK activity via their effectors, Rho-associated protein kinases (ROCK), myotonic dystrophy kinase-related Cdc42-binding kinases (MRCKα), and p21-activated kinases (PAK), PAK1, PAK2, and PAK4. LIMK1 and LIMK2 are activated by phosphorylation of Thr508 and Thr505, respectively [7,8].

LIM kinases are involved in cytoskeleton dynamics by independently remodelling both actin filaments and microtubules. Their most extensively described substrates are members of the actin depolymerising factor/cofilin (ADF/cofilin) family: cofilin1 (non-muscle cofilin, or n-cofilin), cofilin2 (muscle cofilin, or m-cofilin), and destrin (also known as actin depolymerising factor, or ADF), usually regrouped under the term cofilin. Cofilin was discovered as the first substrate of LIMK1 and LIMK2 in 1998 [9,10] and 1999 [7], respectively. Once activated, LIM kinases inactivate the ADF/cofilin proteins by phosphorylating their Ser3, rendering them unable to sever actin polymers and inducing the accumulation of filamentous actin (F-actin), actin stress fibre formation, and impacting cytoskeleton dynamics [7,9,10]. Independently of their activity on the actin cytoskeleton, it has been shown that LIM kinases play a role in microtubule turnover by favouring free tubulin formation [11,12], but the molecular implication of the kinases in this process remains to be elucidated.

As there is growing evidence that LIM kinases are crucial node actors of the cell life fate, in this review, we will recapitulate the role of LIMKs in different cellular events and pathologies, and emphasize the molecular actors involved in these processes. 

## 2. Gene Description and Encoded Proteins

The LIM kinase (LIMK) family of proteins is composed of only two highly related members, LIMK1 and LIMK2, which are encoded by separate genes located on the human chromosomes 7q11.23 and 22q12.2, respectively [13]. There are alternative splicing results in the generation of two *LIMK1* mRNAs: one encodes the full length protein, while the other one results in a truncated protein, missing the beginning of the N-terminal first LIM domain [14]. LIMK2 has three isoforms resulting from this alternative splicing: LIMK2a, LIMK2b, and LIMK2-1 [15,16]. While LIMK2a represents the full-length transcript, LIMK2b lacks half of the first LIM domain. LIMK2-1 differs from its two counterparts with the presence of an extra protein phosphatase 1 inhibitory (PP1i) domain in its C-terminal extremity and a slightly truncated kinase domain [16,17]. A testis-specific LIMK2 isoform, tLIMK2, which lacks LIM domains at the N-terminus due to the usage of a testis-specific alternative initiation exon, is highly expressed in male germ cells and has also been described in mice [18,19]. 

LIMK1 and LIMK2 have the same domain organisation, with two amino-terminal LIM domains, an adjacent PDZ domain, a serine/proline rich region, and a carboxyl-terminal kinase domain (Figure 1). LIMK1 and LIMK2 share a 50% overall sequence identity, and that percentage reaches 70% in the kinase domain [13,20]. The LIM domains (named for the transcription factors Linl1, Isl1, and Mec-3), each composed of two zinc fingers, could play a role in protein–protein interaction, along with the PDZ domain, as well as in protein-to-DNA interaction. The PDZ domain (named for the post-synaptic density protein 95, *Drosophila* disc large tumour suppressor, and Zonula occludens-1 protein), in addition to its protein–protein interaction function, contains two leucine-rich nuclear export signals (NES) and plays a role in nuclear/cytoplasmic shuttling. The C-terminal extremity of LIMKs contains a nuclear localisation sequence (NLS) that could indicate a preferential localisation in the nucleus [21,22]. It also contains an atypical kinase domain that is able to phosphorylate serine and threonine, as well as tyrosine, due to an unusual consensus sequence (DLNSHN motif) in the subdomain VIB of the catalytic site [13].

The structural aspects of LIMK regulation and pharmacology are more extensively described in the review of Chatterjee et al., which belongs to the Special Issue LIM Kinases: From Molecular to Pathological Features [23]. 

## 3. Regulation of LIM Kinases—Partner Network

### 3.1. LIMKs, Downstream Effectors of Small Rho GTPases

LIM kinases were initially identified as kinases that are located downstream of the members of the Rho family of small GTPases, RhoA, Rac1, and Cdc42. Canonically, LIMK1 and LIMK2 are activated by p-21 activated kinases (PAK1,2,4), myotonic dystrophy kinase-related Cdc42-binding kinases (MRCKα), and Rho-associated protein kinases (ROCK1 and ROCK2), via the direct phosphorylation of Thr508 and Thr505, respectively. The phosphorylation of LIMKs on their activation loop leads to an activation of their kinase activity, resulting in a higher amount of phosphorylated cofilin on their Ser3, and its subsequent inactivation and cytoskeleton remodelling [7,8] (Figure 2). LIMKs’ phosphorylation of cofilin is counteracted by phosphatases dephosphorylating phospho-cofilin: SSH (slingshot phosphatases), PP1 (protein phosphatase 1), PP2A (protein phosphatase 2A), and CIN (chronophin).

### 3.2. Other Activators of LIMKs

Other activators of LIMK activity have been reported. They were extensively described in the review of Manetti [20], and we will proceed with an exhaustive update (Figure 3 and Figure 4). 

Protein kinase A (PKA) phosphorylates LIMK1 at Ser596, and, to a lesser extent, at Ser323. As Ser596 is not conserved in LIMK2, it may play a role in the distinct regulation of LIMK1 and LIMK2. Ser596 phosphorylation with PKA increases the LIMK1 kinase activity on cofilin and induces actin cytoskeleton remodelling [24].

Upon VEGF-A (vascular endothelial growth factor A) activation, p38-MAPK (mitogen-activated protein kinase) phosphorylates LIMK1 on its Ser310 without affecting its kinase activity on cofilin. MK2, a MAPKAPK-2 (mitogen-activated protein kinase-activated protein kinase 2) and a downstream kinase of p38-MAPK, also phosphorylates LIMK1, but on its Ser323, leading to an increase in its kinase activity on cofilin [25].

Upon DNA damage, p53 upregulates LIMK2 expression via its binding to an intronic consensus p53 binding site of LIMK2. p53 is a transcription factor that regulates the transcription of various genes that are implicated in cell cycle arrest, autophagy, and apoptosis, upon DNA damages [26]. This regulation is LIMK2-isoform-dependent, as LIMK2b and LIMK2-1, but not LIMK2a, are upregulated, and leads to G2/M arrest via a cofilin phosphorylation increase [26,27]. 

Birkenfeld et al. have shown an interaction between LIMK1 and 14-3-3ζ, a member of the 14-3-3 protein family that is involved in cell signalling, cycle control, and apoptotic death, via yeast two-hybrid screening and GST pull-down experiments [28]. 14-3-3ζ also interacts with cofilin. 14-3-3ζ was shown to preferentially interact with the phosphorylated form of cofilin and to protect it from phosphatases, thus prolonging its inactivation [29]. 

An interaction between LIMK1, but not LIMK2, and p57^Kip2^ (cyclin-dependant kinase inhibitor) was shown through co-immunoprecipitation experiments [30,31]. This interaction is independent of p57^Kip2^ activity and not mediated by ROCK activation. LIMK1–p57^Kip2^ interaction leads to an increase in cofilin phosphorylation and to actin cytoskeleton remodelling. 

LIMK1 was shown to autophosphorylate [6]. This autophosphorylation is rather a transphosphorylation, as it was shown to be promoted by homodimerization that was mediated by Hsp90. Indeed, interactions between LIMK1 and Hsp90, as well as between LIMK2 and Hsp90, were detected via co-immunoprecipitation experiments. A proline plays a crucial role in this interaction. The transphosphorylation of LIMK results in its stabilization and highly increases its lifespan. 

Upon BDNF (brain-derived neurotrophic factor) stimulation, TrkB (Tropomyosin-related kinase B, a tyrosine kinase receptor) was shown to dimerize, leading to LIMK1 dimerization, transphosphorylation, and stabilisation, with a re-localisation from cytoplasm to membrane, resulting in an increased level of phospho-cofilin. TrkB and LIMK1 were shown to interact together by yeast two-hybrid screening and by co-immunoprecipitation experiments. TrkB kinase activity is not required to induce LIMK1 dimerization. LIMK1 was also shown to interact with TrkA and TrkC, two TrkB homologues [32].

LIMK2 activation by Aurora kinase A (AURKA) has also been reported [33]. As AURKA is well known for its implication in cancer development, its role in LIMK2 activation will be more extensively discussed in the breast cancer subsection.

### 3.3. Negative Regulators of LIMKs

LIM kinase activity is finely up- but also down-regulated, and several proteins are able to negatively regulate this (Figure 4). 

Nischarin, a protein involved in intracellular signalling, forms a complex with PAK and LIMK1. Nischarin was shown to interact specifically with the phosphorylated form of LIMK1 on Thr508 via co-immunoprecipitation experiments, leading to its dephosphorylation and inhibition, and resulting in lower amounts of phospho-cofilin and impaired cancer cell invasion [34]. Nischarin acts synergistically with LKB1 (liver kinase B1) to decrease the phosphorylation of PAK, LIMK, and cofilin, reducing stress fibres and inhibiting cell migration and invasion [35].

LATS1 (large tumour suppressor kinase 1) was shown to interact with LIMK1 by co-immunoprecipitation experiments, leading to a lower amount of phosphorylated cofilin. LATS1 and LIMK1 co-localise at the actomyosin contractile ring. The inhibition of LIMK1by LATS1 reverts the LIMK1-induced cytokinesis defects [36].

Slingshot phosphatase (SSH, also known as SSH-1 or SSH1), initially described as cofilin phosphatase, also dephosphorylates and inactivates LIMK1 and LIMK2. SSH was shown to interact with LIMKs by co-immunoprecipitation experiments, and neither the catalytic activity of LIMK2, nor that of SSH, are required for this interaction. SSH dephosphorylates LIMK1 on Thr508, but also on trans/auto-phosphorylated residues, leading to decreased cofilin phosphorylation. A complex SSH1/LIMK1/Actin/14-3-3ζ was identified, and PAK4 was shown to interact with SSH1, leading to its phosphorylation and inhibition [37]. 

Upon PAR-2 (protease activated receptor) activation, a complex between the scaffold β-arrestin, cofilin, chronophin, and LIMK is formed, leading to cofilin dephosphorylation and actin filament severing. The proteins β-arrestin-1 and 2 were shown to interact with LIMK through co-immunoprecipitation experiments. The PAR-2 inhibition of the LIMK activity towards cofilin requires β-arrestin and triggers the LIMK re-localisation to membrane protrusions [38].

The RING finger E3 ubiquitin ligase (Rnf6) is highly expressed in the axons of developing neurons during mouse embryogenesis, and can influence the axon outgrowth of cultured hippocampal neurons. Rnf6 binds to and catalyses the polyubiquitination of LIMK1, which leads to its degradation by the proteasome in growth cones. An interaction between Rnf6 and LIMK1 was shown by co-immunoprecipitation experiments. In the presence of Rnf6, the LIMK1 lifespan is reduced 5-fold (4 h vs. 20 h) [39]. 

Par-3, a member of the polarity proteins involved in the formation of tight junctions, was shown to inhibit LIMK2, but not LIMK1. Par-3 was shown to interact with LIMK2 by co-immunoprecipitation, leading to its inhibition, as phospho-cofilin levels were strongly reduced in its presence [40]. 

### 3.4. Other Partners of LIMKs

LIMKs were also shown to interact with other proteins without phosphorylating them (Figure 4). 

The orphan nuclear receptor Nurr1 was shown to interact with LIMK1 via GST pull-down. LIMK1 inhibits the Nurr1 transcriptional activity, but the molecular requirements of this inhibition have not been elucidated [41]. 

LIMK1 was also shown to interact with paxillin, but not with vinculin, through co-immunoprecipitation experiments, although it colocalizes with actin, paxillin, and vinculin at the focal adhesions in fibroblasts [5]. 

Fascin-1, an actin crosslinking protein, was shown to interact with LIMK1 and LIMK2 by FRET and co-immunoprecipitation experiments. The LIMK phosphorylation by ROCK on Thr505 and Thr508 is required for this interaction, but LIMK kinase activity is not [42].

LIM kinases were also shown to play a pivotal role in the coordination of microtubules and actin filament dynamics. They interact with both actin and tubulin. The ROCK phosphorylation of LIMK1 increases its interaction with actin, whereas it decreases its interaction with tubulin, as was shown by co-immunoprecipitation experiments [11]. LIMK1 overexpression leads to microtubule destabilisation, and its kinase activity is required for this process. LIMK1 was initially thought to phosphorylate TPPP1 (tubulin polymerisation promoting protein 1), but it was then shown that TPPP1 was the substrate of ROCK [43]. Actually, a trimeric complex between LIMK1, TPPP1, and HDAC6 (histone deacetylase 6) was identified. ROCK inhibition stabilises this complex, whereas its activation dissociates it. This trimeric complex leads to HDAC6 inhibition and a subsequent increase in microtubule Lys40 acetylation, which renders microtubules more resilient to mechanical breaks, leading to longer-lived MT [44]. On the contrary, it decreases the cofilin phosphorylation level, resulting in the destabilisation of actin stress fibres. LIMK overexpression leads to tubulin acetylation and microtubule stabilisation, whereas TPPP1 overexpression leads to lower levels of phosphorylated cofilin and stress fibre disruption [45].

## 4. LIMK Substrates

Cofilin was the first substrate of LIMK that was identified [9,10]. The other substrates of LIMKs have been described since then (Figure 4).

The transcription factor cAMP-responsive element binding protein (CREB), a transcription factor that regulates the genes responsible for cell proliferation, differentiation, and survival, is phosphorylated by LIMK1 on its Ser133. An LIMK1–CREB interaction was shown via co-immunoprecipitation experiments [46].

Upon VEGF activation, LIMK1 phosphorylates annexin 1, a calcium- and phospholipid-binding protein, in an in vitro assay (^32^P labelling). This interaction is thought to regulate endothelial cell migration upon VEGF stimulation [47].

A trimeric complex between LIMK, Orb2, and Tob was also described. *Drosophila* Orb2 belongs to the cytoplasmic polyadenylation element binding proteins (CPEB), and binds RNA and regulates translation. Tob (transducer of Erb-B2) is known to induce Orb2 oligomerisation. More recently, it was shown that LIMK phosphorylates Tob, then associates with Tob to phosphorylate Orb2, leading to its stabilisation and oligomerisation. Orb2 oligomerisation plays a major role in long-term memory [48].

The membrane type-1 matrix metalloproteinase (MT1-MMP/MMP14) interacts with LIMK1 and LIMK2 in co-immunoprecipitation experiments, which triggers its phosphorylation on Tyr573. This phosphorylation leads to MT1-MMP-positive endosome association with cortactin, an f-actin binding protein, resulting in invadopodia formation and matrix degradation. LIMK1 and LIMK2 seem to play different roles in this process, as LIMK1 is involved in the cortactin association with MT1-MMP in positive endosomes, while LIMK2 is implicated in invadopodia-associated cortactin [49].

## 5. LIMK Physiological Functions

As cytoskeleton remodelling plays a vital role in the life of the cell, LIM kinases are involved in several physiological processes, including cell migration, cell cycle, apoptosis, and neuronal differentiation, which will be developed in the following paragraphs.

### 5.1. Cell Migration

Cell migration is essential to numerous physiological processes, such as embryogenesis, neuronal development, immune response, and wound repair. Cell morphogenesis during migration requires the fine spatiotemporal remodelling of the cytoskeleton and can be divided into four steps: (i) the protrusion of the leading edge of the cell (filopodia and lamellipodia, resulting from cytoskeleton polymerisation), (ii) the attachment of the cell front via focal adhesions (FA), (iii) the contraction of the whole-cell body through the interaction between myosin and actin in FA-anchored stress fibres, and (iv) the detachment and retraction of the rear by FA disassembly [50]. The implication of the Rho family of small GTPases in cell migration is well documented [51,52,53]. As they are the downstream effectors of small Rho GTPases, LIMKs play a role in this process. Moreover, the initial characterisation of LIMKs as kinases that phosphorylate cofilin pointed out their role in actin cytoskeleton dynamics and cell migration [9,10]. The involvement of LIMKs in microtubule remodelling is another way for them to be involved in cell migration [11].

The role of LIMKs in the polarised migration of immune cells upon chemokine stimulation, i.e., chemotaxis, was particularly emphasized. Indeed, LIM kinases and SSH-1 appear to play a key role in the cofilin-driven assembly and disassembly of protrusions [54,55]. Nishita et al. showed that, upon LIM kinase knockdown, lamellipodium and cell migration are suppressed, while SSH-1-KD cells display an impaired directional cell migration. These authors defined that LIMK1 worked synergistically with SSH-1 to assure the deactivation of cofilin at the leading edge of the chemokine-stimulated Jurkat T cells. At the rear, the activation of cofilin allows for the renewal of G-actin for the actin polymerisation in protrusions [54,55]. LIMK1 is required for cell migration by stimulating the lamellipodium formation in the first stages of the cell response, whereas SSH1 restricts this migration to one direction. Rac activation is required for the LIMK1-mediated SDF-1 (stromal cell-derived factor-1) chemotactic response, but not Rho or Cdc42 [56]. Upon EGF stimulation, LIMK1 was also shown to be involved in actin nucleation at the leading edge and the subsequent lamellipod extension in the metastatic adenocarcinoma cell line [57].

Two other partners of LIMKs were shown to play a role in LIMK-induced cell migration. In response to VEGF, MAPK/MK2/LIMK1 pathway activation leads to annexin 1 phosphorylation and activation, which triggers endothelial cell migration and tubulogenesis. LIMK1 was shown to directly phosphorylate annexin 1 in an [γ^32^P]-ATP labelling on LIMK1 immunoprecipitation, upon VEGF stimulation of human umbilical vein endothelial cells (HUVECs). p38-MAPK is required for this phosphorylation [47]. Fascin-1, an actin crosslinking protein that plays a role in the assembly of cell protrusions, is also associated with a complex of activated LIM kinases, promoting filopodia stabilisation. Indeed, LIMK1/fascin-1 interaction was shown by FRET and FLIM experiments, as well as by His pulldown. LIMK1 activation via ROCK is required for this interaction and for the subsequent filopodia formation and stability [42,56]. Finally, cell migration is dysregulated in cancer, and LIM kinases have been associated with higher tumour invasion properties and metastasis, indicating that the dysregulation of LIMK-mediated cell migration can lead to tumorigenesis [58,59,60,61]. Indeed, many inhibitors targeting LIMK activity have been shown to affect cell migration, as it is described in Berabez et al.’s review, published in the Special Issue LIM Kinases: From Molecular to Pathological Features [62].

### 5.2. Cell Cycle

Cell division is a complex and highly regulated mechanism that requires the fine remodelling of actin filaments and microtubules [63]. As main regulators of cytoskeleton dynamics, LIM kinases are involved in this process. LIMK localisation was extensively studied during the cell cycle and it was shown that LIMKs display different cellular localisation, depending of the stage of the cell cycle of HeLa cells [3]. During interphase and prophase, LIMK1 is associated with cell–cell adhesion sites and re-localises at the spindle poles during metaphase [3]. It disappears from these regions during late anaphase and redistributes at the contractile ring and cleavage furrow during telophase/cytokinesis. During interphase, LIMK2 is diffused throughout the cytoplasm. It re-localises at the spindle pole during prophase and redistributes at the mitotic spindles during metaphase and early anaphase along the spindle microtubules. As the cells progress through late anaphase and telophase, the LIMK2 localises at the spindle midzone, where it was seen to co-localise with microtubules [3]. These observations indicate that LIMK1 and LIMK2 might play different roles during the course of the cell cycle.

LIMK localisation during mitosis was also studied on other cell lines: LIMKs are located at the centrosomes and at the cleavage furrow of MDA-MB-231 breast cancer cells and DU145 prostate cancer cells. Phospho-LIMK, but not LIMK, co-localises with gamma-tubulin in centrosomes. An interaction between phospho-LIMK, but not LIMK, and gamma-tubulin, was detected by co-IP on crude nuclear extracts [64]. SSH1 localisation was also determined in HeLa cells during mitosis. It appears at the actin cortex during metaphase and redistributes at the cleavage furrow during anaphase and telophase [65]. The similar localisation of SSH1 and LIMKs suggests a fine regulation of cofilin phosphorylation/dephosphorylation, and thus, the inactivation/activation progresses through the cell cycle and actin filament remodelling. Furthermore, different studies have shown the transient activation of these proteins along the cell cycle progression [3,66,67]. In synchronised HeLa cells, LIMK1 is hyperphosphorylated and activated during prometaphase and metaphase, gradually returning to the basal levels of the phosphorylation and activity in telophase and during cytokinesis. As for LIMK, cofilin phosphorylation is increased during prometaphase and metaphase, with a gradual return to basal levels during telophase and cytokinesis, indicating that the LIMK1 hyperphosphorylation activated its kinase activity towards cofilin [67]. The activity of LIMK2 did not change after the cells were released from an S phase cell cycle block. However, when the cells were treated with nocodazole or taxol to disrupt the microtubules and induce an M-phase block, LIMK2 was activated, suggesting that LIMK2 might be responsive to a spindle checkpoint [3]. The SSH1 activity decreased during early mitosis and returned to basal levels during telophase and cytokinesis, resulting in cofilin dephosphorylation and activation [65]. LIMK2 hyperphosphorylation during the early stages of mitosis was also observed, but to a lesser extent than that of LIMK1 (1.8-fold versus 6.4). This hyperphosphorylation leads to the enhanced phosphorylation of cofilin. This activation seems to be mediated via the LIM/PDZ domains of LIMK1, as no increased activity on the cofilin by the restricted kinase domain of LIMK1 was observed. Furthermore, LIMK1-Thr508 phosphorylation is required for this process, but not via ROCK or PAK activation [67]. This activation may occur via cyclin-dependent kinase (Cdk), as it is eliminated in the presence of roscovitine, a Cdk inhibitor [66].

LIMK misexpression leads to an aberrant cell cycle progression. Indeed, the inhibition of the LIMK1 activity during mitosis leads to a delay in mitotic progression (metaphase to anaphase) and irregular spindle positioning [68,69,70]. On the contrary, the overexpression of LIMK1 or a phosphatase-inactive SSH1 results in increased levels of phosphorylated cofilin and the production of multi-nucleated cells [67]. Upon ROCK activation in NIH-3T3 mouse fibroblasts, LIMK2, but not LIMK1, induces cyclin A expression and decreases p27^Kip1^ expression, which is a Cdk inhibitor, thus regulating the progression through the G1 to S phase. p57^Kip^, another Cdk inhibitor, was shown to interact with LIMK1, but not LIMK2, by co-immunoprecipitation on HeLa cells. It directly promotes LIMK1 phosphorylation, but not on Thr508, and without going through ROCK. Furthermore, the Cdk inhibitory activity of p57^Kip2^ is not required for its LIMK-mediated kinase activation on cofilin. Therefore, this cell cycle progression control by p57^Kip2^ occurs via its direct activation of LIMK1 [30].

Furthermore, LIMK1, but not LIMK2, was shown to interact with LIC1 and LIC2 (dynein light chain) via its kinase domain, through co-immunoprecipitation. LIMK1 promotes LIC1 and LIC2 phosphorylation on tyrosine residues. LIC1 and LIC2 are two subunits of the huge complex that forms the cytoplasmic dynein 1 motor, and defines the cargo specificity of dynein. LIMK1 seems to affect dynein motor function, as the kinase dead mutant of LIMK1, D460A, reduces the speed of PLK1 trafficking (a cargo of the dynein motor). Furthermore, LIC1 and LIC2 rescued the aberrant cell cycle progression induced by LIMK1 depletion (multipolar spindle, centrosome spread length, and spindle pole density). LIMK1 could regulate the trafficking of pericentriolar proteins by dynein cargo transportation via LIC1 and LIC2 phosphorylation [69].

### 5.3. Apoptosis

Apoptosis is a mechanism of tightly regulated programmed cell death, and an essential process in development and cell homeostasis. Apoptosis may be induced by extrinsic stimuli (death receptor pathway) or intrinsic stimuli (mitochondrial pathway), leading to caspase proteolytic pathway activation [71]. A drastic change in cell morphology occurs during apoptosis: shrinkage, bleb formation, and rounding-up. The actin cytoskeleton has been linked to these apoptotic phenotypes [72].

Fas receptor, also known as Fas, FasR, apoptosis antigen 1 (APO-1 or APT), cluster of differentiation 95 (CD95), or tumour necrosis factor receptor superfamily member 6 (TNFRSF6), is a cell death surface receptor. The treatment of Jurkat and HeLa cells with an anti-Fas antibody leads to the cleavage and activation of LIMK1. The produced LIMK1 N-terminally truncated fragments are constitutively active and stimulate membrane blebbing when they are overexpressed, ultimately leading to cell death. The pre-treatment of cells with benzyloxycarbonyl-Asp(OCH_3_)-Glu(OCH_3_)-Val-Asp(OCH_3_)-fluoromethylketone (z-DEVD-fmk), an inhibitor for caspase-3 or related proteases, blocked the appearance of these LIMK1 fragments, as well as LIMK1 activation, suggesting that the cleavage of LIMK1 is mediated by these caspases. Indeed, LIMK1 is cleaved at a short motif DEID within the PDZ domain, more precisely at aspartic acid 240, while the caspase-3 and related proteins cleave the protein at a DEXD site. Interestingly, LIMK2 does not possess the DEID motif, suggesting that the cleavage-mediated LIMK activation is LIMK1-specific. Moreover, LIMK1 silencing using siRNA partially suppresses membrane blebbing. The caspase-mediated specific cleavage and activation of LIMK1 might play a decisive role in the membrane bleb formation during apoptosis [73].

Apoptosis is induced in the prostate cancer cells LNCaP (hormone sensitive), as well as in DU145 (hormone insensitive), when stimulated with membrane androgen receptor agonist testosterone-BSA (a non-permeable steroid albumin conjugate). In these conditions, Rho, ROCK, and LIMK2 are activated. LIMK2 immunoprecipitated extracts from the testosterone-BSA-treated DU145 exhibit a higher kinase activity compared to untreated cells, with an increase in phospho-LIMK and phospho-destrine by [γ^32^P]-ATP labelling. This increase, as well as apoptosis induction, is lost when cells are pre-treated with the ROCK inhibitor Y27632 [74].

Apoptosis is also induced in the colon cell line HCT116 upon tumour necrosis factor (TNF) treatment. In these conditions, an increase in LIMK phosphorylation on its Thr508 and the subsequent phosphorylation of cofilin are observed. Death-associated protein kinase (DAPK), a protein implicated in programmed cell death, could act as a scaffold protein for the LIMK/cofilin complex in TNF-induced apoptosis. The DAPK/LIMK/cofilin complex is abrogated once the cells are committed to apoptosis [75]. Indeed, LIMK and cofilin are co-immunoprecipitated with DAPK in extracts from the cells treated with TNF. Thus, DAPK promotes a closer interaction between LIMK and cofilin, resulting in a higher phosphorylation of cofilin by LIMK. DAPK kinase activity is required for this increase, as it is lost when the cells are pre-treated with a DAPK inhibitor. Furthermore, TNF induces the re-localisation of the DAPK/LIMK/cofilin complex in the perinuclear compartment.

p57^Kip2^ has been shown to enhance mitochondrial-mediated apoptosis via the activation of LIMK1 and the stabilisation of the actin cytoskeleton. It has been shown to interact with LIMK1, leading to LIMK1 activation in HeLa cells [30]. Apoptosis triggered by staurosporine, an alkaloid with antibiotic properties, is mediated by LIMK1 in the HeLa cells overexpressing p57^Kip2^. Indeed, in these conditions, the silencing of LIMK1 leads to a decrease in apoptotic nuclei, Cas3 activity, and PARP cleavage. Furthermore, hexokinase-1, an inhibitor of the mitochondrial voltage-dependent anion channel, is displaced from mitochondria, inducing mitochondrial depolarisation and apoptotic cell death [76].

Upon a genotoxic stress, increases in RhoGTP, phospho-LIMK, and phospho-cofilin are observed, as well as cytoskeleton rearrangement with cell flattening and enhanced stress fibres. An increase in LIMK2-1 and LIMK2b, but not LIMK2a, at the protein and mRNA levels is also observed, requiring p53 transcriptional activity and leading to cell survival. LIMK2 inhibition was shown to sensitise cells to DNA-damage-induced apoptosis [27]. Hence, the LIMK inhibitors associated with genotoxic compounds could constitute an efficient alternative therapy to treat the cancer cells that are resistant to chemotherapy. Indeed, LIMK2 has been shown to be mis-regulated in these resistant cells [12,77].

LIMK2 is also involved in a particular form of neuronal cell death (necrosis) that results from status epilepticus (SE). SE is defined by a single long-lasting or multiple successive life-threatening seizures. Kim et al. have shown that the neuronal death in rat CA1 neurons is driven by a cyclin D1-CDK4 complex upon SE, and that this complex leads to the overexpression of LIMK2. When overexpressed, the LIMK2 impairs dynamic-related protein-1 (DRP1)-mediated mitochondrial fission by stabilising F-actin and preventing DRP1-actin interaction. Thus, it induces mitochondrial elongation and neurotoxicity. LIMK2 silencing via siRNA prevents the downregulation of the DRP1 and mitochondrial elongation induced by SE [78]. These authors further elucidated the stimulus leading to this LIMK2-mediated neuronal necrosis in an SE context. ET-1, a vasoactive peptide produced by vascular endothelium, and its brain receptor ET_B_, are rapidly increased upon SE. BQ788, an ET_B_ antagonist, diminished the SE-induced neuronal damage via the alleviation of ROCK1 upregulation and a reduction in the LIMK2 protein and mRNA expression [79]. A TE-1 injection into the hippocampus of a normal rat leads to an increase in the LIMK2 mediated by ET_B_, as well as in mitochondria elongation and sphere formation, confirming the data obtained upon SE.

### 5.4. Neurodevelopment and Synaptic Plasticity

LIMKs have been shown to play a crucial role during development. Ribba et al. notably focus on embryonic development in their review, which was published in the Special Issue LIM Kinases: From Molecular to Pathological Features [80]. Here, we choose to concentrate on neurodevelopment, synaptic plasticity, memory, and brain functions, as they are among the most well described functions of LIM kinases, along with their function in cytoskeleton remodelling. These features have been extensively detailed in several exhaustive reviews [81,82,83]. We will focus on the most striking data, especially that in KO mouse models.

Very quickly after their discovery, the implication of LIMKs in neurodevelopment was pointed out in knockout (KO) mice. Indeed, LIMK1 seems to be involved in dendritic spine regulation, because LIMK1 KO mice neurons exhibit abnormal dendritic spines, which are longer, thinner, and immature. Accordingly, the spine heads of these LIMK1 KO neurons display a reduced amount of actin filaments, indicating that LIMK1 plays a critical role in spine morphology through actin filament remodelling. Indeed, the phospho-cofilin levels were higher in LIMK1 KO mice neurons [84]. LIMK2 also seems to play a role in spine regulation. Although LIMK2 KO mice have mild synaptic dysfunction, LIMK1 and LIMK2 double KO mice exhibited aggravated effects compared to that of the LIMK1 KO on the synaptic function, suggesting that there might be a compensatory mechanism upon the loss of LIMK1 or LIMK2 [85]. Several studies suggest that PAKs, as well as ROCK2, could be the upstream activators of LIMKs, with regard to their function in spine regulation [86,87,88].

There is evidence that LIM kinases also play a role in long-term potentiation (LTP) and long-term depression (LTD), two mechanisms that underlie synaptic plasticity, learning, and memory processes. Meng et al. showed that LIMK1 KO mice exhibited enhanced hippocampal LTP, as well as altered fear responses and spatial learning. F-actin depolymerising toxins increased the early-phase LTP (E-LTP) in wild-type mice, an effect that was abolished in the LIMK1 KO mice, suggesting that the role of LIMK1 on E-LTP is mediated by the actin cytoskeleton [84]. In 2015, Todorovski et al. showed that LIMK1 KO mice were drastically impaired in long-term memory (LTM) but not short-term memory (STM), and were defective in late-phase long-term potentiation (L-LTP), a form of long-lasting synaptic plasticity that is specifically required for the formation of LTM. They also showed that L-LTP-deficient mice were rescued by a pharmacological increase in the cAMP response element binding protein (CREB) activity [89]. CREB, as a transcription factor that regulates the genes responsible for cell proliferation, differentiation, and survival, is critical for LTM establishment. LIMK1 and CREB were shown to interact by two-hybrid screening and co-IP experiments, as well as in different parts of the brain [46], and LIMK1 has been shown to phosphorylate CREB [89].

### 5.5. Membrane Trafficking

Membrane trafficking is a complex process that allows for the delivery of specific cargo (proteins and macromolecules) via transport vesicles towards the dedicated location. LIMK1 has been shown to be involved in membrane trafficking from the Golgi apparatus.

Rosso et al. showed that LIMK1 is enriched in the Golgi apparatus of developing neurons, with its LIM domain triggering this localisation. LIMK1 seems to regulate Golgi dynamics, as it is involved in the tubule-vesicular process. Indeed, when overexpressed, LIMK1 abrogates the formation of trans-Golgi-derived tubules and prevents cytochalasin D-induced Golgi fragmentation, as well as the Golgi export of synaptophysin-containing vesicles. LIMK1 kinase dead mutant has the opposite effect. LIMK1 enhances the accumulation of Par3/Par6, insulin-like growth factor 1 (IGF-1) receptors, and the neural cell adhesion molecule (NCAM) at growth cones, suggesting that it plays a crucial role in the delivery of these proteins to growth cones. These results suggest that a key role is played by LIMK1 in the Golgi dynamics and membrane trafficking in neurons [90]. Salvarezza et al. also reported the implication of LIMK1 and cofilin in the trafficking of proteins out of the Golgi apparatus in Madin–Darby canine kidney cells (MDCK). They showed that LIMK1, but not LIMK2, is localised at the Golgi apparatus, where it induces a specialised population of actin filament apparatus that is required for the emergence of an apical cargo route to the plasma membrane (PM), with a high specificity. Indeed, LIMK1, but not LIMK2, regulates the exit from the trans-Golgi network (TGN) of the apical PM marker p75 neurotrophin receptor and NHR2, a related receptor, but is not involved in the exit of another apical PM marker glycosyl phosphatidylinositol (GPI), nor the basolateral PM marker neural vell adhesion molecule (NCAM). The overexpression of a kinase dead LIMK1 mutant, a constitutively activated cofilin, or the use of LIMK1 siRNA, selectively slowed down this exit from the TGN. These authors also showed that in p75 carrier vesicles, LIMK1 cooperates with dynamin 2, and cortactin and syndapin, two dynamin-interacting proteins for the fission processes of the vesicle from the TGN [91]. To our knowledge, the role of LIMK1 in membrane trafficking was only documented by these two papers.

## 6. Involvement of LIMKs in Different Pathologies

### 6.1. Cancer

Because they play a crucial role in actin cytoskeleton remodelling, cell shape, proliferation, and motility, there is growing evidence that LIM kinases are involved in tumour cell invasion, tumour growth, and metastasis. Indeed, LIMK1 and LIMK2 have been shown to be upregulated in breast cancer [58], gastric cancer [59], prostate cancer [92,93], and malignant melanoma cells [94], and they seem to be involved in multiple non-canonical signalling pathways that, when dysregulated, actively participate in tumorigenesis.

#### 6.1.1. Breast Cancer

With 2.3 million diagnoses in 2020, breast cancer is the most common and deadliest form of cancer in women [95]. LIMK1 and LIMK2 have been reported to be overexpressed in breast cancer [58,96], and there is growing evidence of their implication in molecular pathways with interactors linked to breast cancer tumorigenesis (Figure 5).

Aurora kinase A (AURKA) is overexpressed in several types of cancer, including breast cancers, where it plays a role by aberrantly phosphorylating the proteins implicated in the cell cycle, ultimately leading to cell malignant transformation [97]. LIMK2 directly interacts with AURKA via its LIM domains in coIP experiments, which increases its levels by inhibiting its ubiquitin-dependent degradation and restraining its localisation to the cytoplasm. AURKA is also able to activate LIMK2 by the phosphorylation of its Ser283, Thr494, and Thr505, resulting in: (i) the increased kinase activity on cofilin, (ii) higher levels of LIMK2 via the prevention of its ubiquitin-dependent degradation, and (iii) the restraining of its localisation in the cytoplasm. Reciprocally, LIMK2 is required for AURKA-mediated cellular transformation in breast cancer, indicating that LIMK2 is a key oncogenic effector of AURKA in breast cancer malignancy [33].

Serine-arginine protein kinase 1 (SRPK1) has also been described as a target of LIMK2, and this connection plays a major role in triple negative breast cancer metastasis. Through SILAC experiments that used LX7101, an inhibitor of LIMKs, Malvi et al. have shown a strong decrease in SRPK1 phosphorylation on several serines (Ser7, Ser9, Ser51, Ser309, and Ser311) [96]. SRPK1 is involved in the splicing of pre-messenger mRNA, a mechanism which is imbalanced in malignant tumour cells. When overexpressed, SPRK1 is responsible for resistance to apoptotic signals [98], resistance to cisplatin therapy [99], and enhanced metastasis [100]. LIMK2 is overexpressed in triple-negative breast cancer (TNBC) and pharmacological inhibition, with LX7101 or TH-257, leads to the inhibition of the metastatic characteristics of TNBC cells (migration, invasion, actomyosin contractility, and extracellular matrix degradation). No direct interaction by coIP was detected between SRPK1 and LIMK2. However, an in vitro kinase assay pointed out the serine phosphorylation of SPRK1 on a recombinant protein, validated by immunoblotting. The inhibition of SRPK1 blocked the metastatic properties of TNBC cells, which led to the thought that LIMK2 promotes the metastatic progression of triple-negative breast cancer by activating SRPK1. The pharmacological inhibition of LIMKs by LX7101 inhibits metastatic progression in mice, but has no effect on primary tumour growth [96]. As LX7101 is a dual inhibitor of LIMKs and their upstream regulating kinase ROCK, we cannot rule out a role of ROCK in this phenomenon. However, another assay with the LIMK inhibitor Pyr1 on xenografted mice that were developing breast tumours and paclitaxel resistance showed a blockage of primary tumour growth, but not of their spread [101]. Pyr1 prevents cofilin phosphorylation by inhibiting LIMK1 and LIMK2 in vitro and in cellulo on HeLa cells. Pyr1 is a selective inhibitor of LIMK1: on a panel of 66 kinases, there were only three hits (MLK1, NEK11, and LIMK1), with the highest inhibition observed for LIMK1 (4% residual in vitro kinase activity). A thermal stability shift assay confirmed Pyr1 selectivity for LIMK1 [70].

LIMK1 and LIMK2 were also shown to interact with the membrane-anchored type-1 matrix metalloproteinase (MT1-MMP, or MMP14) in triple-negative breast cancer. To be able to spread, cancer cells need to degrade the extracellular matrix (ECM). In breast carcinoma, the dissemination of metastasis involves MT1-MMP, which participates in the degradation of ECM and is overexpressed in several cancers. A direct interaction between LIMK1/LIMK2 and MT1-MMP has been shown by coIP via the “DVK” motif of MT1-MMP. MT1-MMP phosphorylation by LIMKs on Tyr573 modulates its interaction with cortactin, a F-actin-binding protein. Lagoutte et al. showed that LIMK1 regulates the cortactin association with MT1-MMP-positive endosomes and associates with the MT1-MMP in endosomes, while LIMK2 participates in the formation of invadopodia-associated cortactin pools, indicating that both are necessary for MT1-MMP-induced matrix degradation and the tumour cell invasion in breast tumour [49].

#### 6.1.2. Prostate Cancer

Prostate cancer is the second most common cancer in men after lung cancer [102]. LIMK1 [92] and LIMK2 [93] are overexpressed in prostate cancer and prostate cell lines, and numerous LIMK partners that are dysregulated in prostate cancer have been discovered over the last years.

The group of Shah has been particularly active in demonstrating the role played by LIMK2 in prostate cancer over the last years. They pointed out several proteins that are directly phosphorylated by LIMK2, and then ubiquitinated and subsequently degraded by the proteasome. These different partners also regulate the LIMK2 stability by promoting its ubiquitination. Many feedbacks loops are described [93,103,104,105] (Figure 6).

Nikhil et al. depicted Twist-related protein 1 (TWIST1) as a new partner of LIMK2. TWIST1 is a transcription factor involved in embryonic development and organogenesis [93]. It is overexpressed in many cancers, where it drives tumour initiation, angiogenesis, dissemination, and drug resistance [106]. In adults, TWIST1 expression is limited to the quiescent stem cells located in mesenchymal tissues, but it is upregulated following androgen deprivation therapy (ADT) via TGF-β signalling, mediating the cancer prostate aggressiveness and development of castration-resistant prostate cancer (CRPC), a metastatic form of prostate cancer. LIMK2 and TWIST1 are interconnected in a synergic feedback loop. In an in vitro kinase assay with purified proteins, the authors show that LIMK2 phosphorylates TWIST1 on four different serines (Ser45, Ser78, Ser95, and Ser199), resulting in a decreased level of the ubiquitination of TWIST1 and, thereby, its stabilisation. This TWIST1 phosphorylation by LIMK2 is required for cell growth and migration, promotes epithelial–mesenchymal transition (EMT), and is crucial for tumorigenesis in vivo in xenograft mice. TWIST1 phosphorylation and stabilisation via LIMK2 also increases LIMK2 levels by inhibiting its ubiquitination and thus, its degradation. LIMK2 silencing decreases the TWIST1 mRNA levels under hypoxic conditions, but not under normoxia, whereas it decreases the TWIST1 protein levels under both conditions. Furthermore, the sequential treatment of prostate cancer cell line C4-2 with docetaxel, a well-established treatment for CRPC, and then with an LIMK2-specific inhibitor [107], resulted in a significant cellular death [93]. This high synergy is very promising for the future treatment of hormone-independent CRPC.

SPOP, an E3 ubiquitin ligase adapter that is involved in numerous cellular mechanisms, is another partner of LIMK2. SPOP is the most mutated gene in CRPC, and a vast majority of these identified mutations alter its ability to ubiquitinate some of its oncogenic targets [108]. No direct interaction between LIMK2 and SPOP was detected by coIP with endogenous proteins. However, in an in vitro assay on purified proteins, SPOP is directly phosphorylated by LIMK2 on Ser59, Ser171, and Ser226, which causes its retention in the nucleus, and decreases its stability by increasing its ubiquitination. SPOP phosphorylation with LIMK2 decreases its ability to: (i) degrade its targets cMyc, androgen receptor (AR), and androgen receptor splice variant-7 (Arv7) by ubiquitination, (ii) inhibit cell proliferation and migration, (iii) prevent tumorigenesis in mice, and (iv) reduce EMT in vivo. Conversely, SPOP targets LIMK2 for ubiquitination and participates in its degradation via the proteasome [103]. Hence, LIMK2 and SPOP are part of a double negative feedback loop. SPOP also interacts with AURKA through coIP on endogenous proteins, which directly phosphorylates it on three sites (Ser33, Thr56, and Ser105), causing its ubiquitination and degradation. SPOP also degrades AURKA via a feedback loop [104]. The positive feedback loop existing between AURKA and LIMK2 [33] could be an aggravating factor to prevent the SPOP tumorigenesis inhibition.

Upon hypoxia, in 22Rv1, PC-3, and LN95 prostate cancer cell lines, LIMK2 is upregulated and phosphorylates the phosphatase and TENsin homolog protein (PTEN) at five sites (Ser207, Ser226, Ser360, Ser361, and Ser362), causing its ubiquitination and degradation. No direct interaction via coIP was reported, but an in vitro kinase assay with purified proteins depicts a direct phosphorylation of PTEN by LIMK2. PTEN is a phosphatase involved in the regulation of the cell cycle. When dysregulated because of gene mutations or post-translational modifications, PTEN will drive the development of CRPC. PTEN is also known to be downregulated in hypoxic tumours, following castration via androgen deprivation therapy (ADT). LIMK2 also inhibits its lipid phosphatase activity upon phosphorylation. They are engaged in a negative regulatory loop, since PTEN promotes the degradation of LIMK2 by ubiquitination as well [109].

Homeobox protein NKX-3.1 is a prostate-specific transcription factor and tumour suppressor protein, which plays a role in cell differentiation, maintenance, and lineage plasticity. Its genetic loss is strongly associated with prostate cancer development. Encoded by an androgen-responsive gene, NKX-3.1 is downregulated following ADT and the subsequent hypoxia. In an in vitro assay using purified proteins, Sooreshjani et al. showed that NKX-3.1 is phosphorylated by LIMK2 on its Ser185. This phosphorylation is required for NKX-3.1 ubiquitination and its subsequent degradation, leading to enhanced cellular growth and migration. Moreover, LIMK2 also regulates NKX-3.1 at the mRNA level. NKX-3.1, in return, promotes LIMK2 ubiquitination and degradation, linking both proteins in a negative feedback loop [105]. Bowen et al. also established a link between NKX-3.1 and PTEN, where PTEN dephosphorylates NKX-3.1 at Ser185, thus enhancing its stability [110]. PTEN, which is downregulated in prostate cancer, has been shown to be an LIMK2 substrate which phosphorylates it and participates in its degradation [109]. Thus, the relations existing between LIMK2, PTEN, and NKX-3.1 could be an aggravating factor in CRPC development.

The LIMK2 targets described above are tumour suppressor proteins, whose degradation upon ubiquitination is promoted by LIMK2 phosphorylation, and is a major step in tumorigenesis progression.

LIMK1, which is found to be upregulated in prostate cancer samples and cancer cell lines, also seems to play a role in prostate cancer pathogenesis [92]. Mardilovich et al. depicted a correlation between elevated LIMK1 expression and activity, high phospho-cofilin levels in prostate cancer patient samples, and poor survival in non-metastatic prostate cancer. LIMK2 expression in non-metastatic PC showed a similar trend, but further investigations are needed. However, LIMK inhibition with a selective LIMK inhibitor, LIMKi 3 (BMS-5), reduced cell motility, inhibited proliferation, and increased apoptosis in androgen-dependent PC cells more effectively than in androgen-independent PC cells. Indeed, LIMKi 3 (BMS-5) has an inhibitory effect on AR nuclear translocation and AR-αTubulin interaction, leading to its retention in the cytoplasm and its subsequent degradation. Since LIMKs are well-described effectors of cytoskeleton remodelling as they are implicated in both actin filament and microtubule rearrangements, an involvement of LIMKs in AR regulation is probable [111]. Hepatocyte growth factor (HGF) has also been linked to the enhancement of prostate carcinoma cell proliferation and invasiveness [112,113]. Ahmed et al. showed that HGF, to which PC-3 prostate cancer cells respond in a chemotactic way, was able to activate PAK4. This activation leads to a specific HGF-driven PAK4/LIMK1 interaction, LIMK1 phosphorylation and activation, cofilin phosphorylation, and an increase in cell motility. Thus, the interaction between PAK4 and LIMK1 could be an essential factor in prostate cancer invasiveness [114].

#### 6.1.3. Leukaemia

The term “leukaemia” regroups different kinds of malignant disorders of the blood and bone marrow. They are characterised by an abnormally high leucocyte count in blood and/or bone marrow and are subdivided in multiple subtypes [115]. The most common one is acute myeloid leukaemia (AML), an aggressive form of leukaemia for which there are very few therapies. LIMK1 and LIMK2 have been described as potential targets for treating AML.

In the human acute monocytic leukaemia cell line THP-1, an interaction between PKCζ and LIMK1, but not LIMK2, was detected by coIP experiments upon CSF1 stimulation [116]. CSF-1 cytokine causes a hematopoietic stem cell differentiation into macrophages. PKCζ, an atypical PKC, is required in chemokine-triggered cell adhesion and the actin assembly in polymorphonuclear cells. Furthermore, the KO of PKCζ by siRNA blocked the CSF-1-induced LIMK1 and cofilin phosphorylation in THP-1 cell lines, resulting in impaired migration. All these data suggest that PKCζ may phosphorylate LIMK1 upon CSF-1 stimulation. LIMK1 might trigger the chemoattraction of macrophages by tumour cells upon CSF-1 release and PKCζ activation, leading to tumour invasion and metastasis [116] (Figure 7).

Jensen et al. performed an RNA interference screening on three different AML cell lines (U937, U60 and OCI-AML3). They identified *LIMK1* as the only gene whose transcripts reduced the cell viability in all these cell lines when targeted by four independent shRNAs. Furthermore, by analysing The Cancer Genome Atlas (TCGA) AML databank, they pointed out a significant association between the high expression of LIMK1 and a shorter survival, and data from the Microarray Innovations in Leukaemia and TCGA AML allowed them to correlate the high LIMK1 mRNA level with the normal karyotype and *KMT2A*-rearrangement AML. The *KMT2A* gene encodes for the histone-lysine *N*-methyltransferase 2A, which works as a positive regulator of gene transcription. As these genetic subtypes are well characterised for recurrent driver alterations, authors have established a significant correlation between a high LIMK1 expression, *FLT3* and *NMP1* mutations, and *KMT2A*-rearrangements by further analysing TCGA AML databank. Suppression by shRNA or KO by the CRISPR-Cas9 gene-editing tool of LIMK1 or LIMK2 reduced the colony formation, decreased proliferation, and induced the morphological changes of different AML cell lines and patient-derived xenograft (PDX) samples, indicating a role for LIMKs in leukaemia tumorigenesis. The suppression of LIMK1 by shRNA was also associated with the upregulation of several tumour suppressor genes (*EGR1*, *BTG2,* and *BIN1*) and the downregulation of mitosis-associated genes (*HOXA9*-associated genes, which are transcription factors). Finally, the authors showed a negative correlation between the LIM kinases and cell division protein kinase 6 (CDK6). CDK6 depletion via shRNA or pharmacological kinase inhibition by Palbociclib leads to a higher level of LIMK1, as well as a higher activity of LIMK1, with an increase in cofilin phosphorylation, and the LIMK1 and LIMK2 suppression leads to an increased CDK6 expression. This negative correlation between LIMKs and CDK6 may be detrimental upon the pharmacological inhibition of both proteins. Authors have shown that the combined inhibition of both CDK6 and LIMKs by Palbociclib and LIMKi 3 (BMS-5), respectively, leads to synergic effects, with decreased proliferation and differentiated morphology [117].

The pharmacological inhibition of LIMKs represents a good therapeutic opportunity in AML. LIMKi 3 (BMS-5), an LIMK inhibitor [118], selectively suppressed the growth of T cell leukaemia (Jurkat and ATN-1), and induced centrosome fragmentation and apoptosis, whereas it had no impact on peripheral blood mononuclear cells [119]. The pharmacological inhibition of LIMKs does appear to be efficient for leukaemia treatment as Pyr1, a selective potent inhibitor of LIMKs, induced a complete survival gain of B6D2F1 mice bearing leukaemia L1210 cells, with no apparent toxicity [70]. Djamai et al. investigated the therapeutic potential of the LIMK1/2 inhibitor CEL_Amide (LIMKi) in FLT3-ITD-mutated (FLT3-ITD+) acute myeloid leukemia (AML). Treatment with LIMKi decreased the LIMK1 protein levels and phosphorylation of cofilin in FLT3-ITD+, MOLM-13, and MV-4-11 cell lines. In MOLM-13 cells, a synergistic effect was obtained when LIMKi was used concomitantly with the FLT3 inhibitors midostaurin, crenolanib, or gilteritinib, while combination experiments with LIMKi and the FLT3 inhibitor quizartinib, hypomethylating agent azacytidine, or ROCK inhibitor fasudil, were additive. Moreover, in NOD-SCID mice that were engrafted with MOLM13-LUC cells, the FLT3 inhibitor midostaurin, or LIMKi alone, delayed the MOLM13-LUC engraftment, and their combination significantly prolonged the survival of leukemic mice. Taken together, these data suggest that the small molecule inhibitor CEL_Amide LIMKi might constitute a novel treatment strategy for FLT3-ITD+ AML, when used in combination with FLT3 inhibitors [120]. The same group evaluated the efficiency of CEL_Amide LIMKi in Philadelphia chromosome-positive (BCR::ABL+) acute lymphoblastic leukemia (ALL), another subtype of leukaemia. In Ph+ (BCR::ABL+) B-ALL, ROCK is constitutively activated, leading to LIMK phosphorylation and thereby cofilin inactivation, and the subsequent abrogation of its apoptosis-promoting activity. In BCR::ABL+ TOM-1 and BV-173 cell lines, and in patient cells, the LIMKi treatment decreased the LIMK1 protein expression, whereas the LIMK2 protein expression was unaffected. Cofilin dephosphorylation was nevertheless observed. The conjoint treatments of CEL_Amide and the BCR::ABL tyrosine kinase inhibitors (TKIs) imatinib, dasatinib, nilotinib, and ponatinib were synergistic for both TOM-1 and BV-173 cell lines. Mice transplanted with CDKN2Ako/BCR::ABL1+ B-ALL cells displayed a prolonged survival when treated with a combination of LIMKi and TKIs, indicating that CEL-Amide might be a promising new therapy for BCR::ABL+ ALL [121].

#### 6.1.4. Osteosarcoma

A comprehensive overview of the role of LIMKs in osteosarcoma is depicted in the review by Brion et al. [122] in this Special Issue, and we will focus here on the main molecular features.

Osteosarcoma (OS) is the most common solid bone malignancy in children and adolescents, characterised by the proliferation of malignant mesenchymal cells which produce osteoid and/or immature bone. Lung metastasis is a frequent evolution of the disease and participates in its poor prognosis [123]. Both LIMKs have been shown to play a role in OS malignancy.

Several studies have shown that LIMK1 is overexpressed in osteosarcoma through immunohistochemistry on patient tissues. A WB analysis and RTqPCR on different OS cell lines (MG63, U2OS, OS732, and SaOS-2) show the overexpression of LIMK proteins and mRNA, respectively, compared to normal osteoblasts (hFOB 1.19) [124,125,126]. This is correlated with higher migration and invasion propensities, and limited apoptosis. Li et al. have shown that the signalling cascade PAK4/LIMK1/cofilin is activated in OS and associated with metastasis and poor survival [125]. PAK4 knockdown or silencing in MG63 diminishes the cell viability, migration, and invasion, and limits the growth of subcutaneous transplanted tumour in nude mice [125].

Different peptides or proteins (Insulin, EGF, and VEGF) have been shown to promote OS proliferation, migration, and invasion. In trying to understand the molecular mechanism involved in these phenomena, the implication of LIMKs has been pointed out. Insulin promotes the proliferation of MG63 osteosarcoma cells in a time- and dose-dependent manner, and also induces LIMK1 and cofilin phosphorylation. Moreover, upon LIMK1 KO using shRNA, insulin-induced proliferation is significantly inhibited, as well as when the cells were treated with the PI3K inhibitor LY294002. In these conditions, LIMK1 phosphorylation is reduced, indicating a possible role for LIMK1 in regulating the osteosarcoma cell proliferation via the insulin/PI3K/LIMK1 signalling pathway [124]. The epidermal growth factor (EGF) also promotes MG63 osteosarcoma cell migration and invasion, as well as stress fibre formation via RhoA activation. All the proteins of the signal transduction pathway RhoA/ROCK1/LIMK2/cofilin are activated upon the EGF treatment of MG63 cells. Moreover, the selective inhibition of ROCK1, LIMK2, or cofilin in the MG63 cells by shRNA prevents actin stress fibre formation and cell migration. These data delineate a role for RhoA/ROCK1/LIMK2/cofilin signalling in actin microfilament formation, and the migration and invasion increase in the MG63 cells upon EGF activation [127]. OS cell metastasis in lungs is an early stage of the pathology, and VEGFR2 and PD-L2 are overexpressed in lung metastasis. VEGFR2 inhibition by shRNA or by apatinib, a small molecular tyrosine kinase inhibitor, reduced the osteosarcoma cell migration, invasion, and metastatic potential, and downregulated the STAT3 and RhoA/ROCK/LIMK2 pathways. STAT3 silencing decreased the LIMK and cofilin phosphorylation. Moreover, VEGFR2 inhibition by apatinibled reduced the PD-L2 expression in osteosarcoma cells and attenuated the osteosarcoma lung metastasis capacity in vivo [128,129,130]. Accordingly, Ren et al. showed that PD-L2 knockdown attenuated the migration and invasion of OS cells, and decreased the LIMK and cofilin phosphorylation, as well as RhoA activation. It also suppressed EMT and inhibited autophagy by decreasing the beclin-1 expression. Beclin-1 knockdown also led to the inactivation of RhoA, as well as an LIMK2 and cofilin phosphorylation decrease, while PD-L2 knockdown inhibited the OS cell metastasis in lungs but had no effect on the primary tumour size [128]. All these data show that VEGFR2 and PD-L2 promote OS lung metastasis via RhoA/ROCK/LIMK2/cofilin signalling pathways. Bone morphogenetic protein type II receptor (BMPR2) was found upregulated in a majority of the osteosarcoma tissues, and its overexpression has been correlated to a poor overall survival. The depletion of BMPR2 in 143B cells diminished the LIMK and cofilin phosphorylation, as well as the cell migration and invasion in vitro, and increased the EMT. In vivo, BMPR2 depletion has no impact on the primary tumour size, however, it inhibits lung metastasis. BMPR2 was shown to interact with LIMK2 by co-immunoprecipitation experiments on U2OS cells. BMPR2 depletion decreased the LIMK and cofilin phosphorylation, whereas BMPR2 overexpression increased the LIMK and cofilin phosphorylation and activated RhoA by promoting its GTP-bound form [131]. Another study has shown an interaction between LIMK1 and BMPR2 by a yeast two-hybrid screening and by the co-immunoprecipitation of the proteins overexpressed in COS7 (via their LIM and C-terminal domain, respectively), or of the endogenous proteins in fibroblasts. An in vitro phosphorylation assay suggested that BMPR2 inhibited cofilin phosphorylation via LIMK1. Furthermore, it was shown that BMPR2 and PAK4 compete to interact with LIMK1 [132]. The discrepancy between these two studies may be explained by different points: (i) LIMK1 and LIMK2 may behave differently towards BMPR2, and (ii) in OS cells, BMPR2 may interact with another partner, modulating its activity towards LIMKs.

Drug resistance is a major problem in osteosarcoma treatment, leading to poor prognosis. Vincristine (VCR) is widely used and is an effective chemotherapeutic agent to treat osteosarcoma. Different studies have explored the molecular mechanism of drug resistance by developing different MG63 cell lines that are resistant to VCR, MG63/VCR [61,126,133]. MG63/VCR appeared to be resistant to many other anticancer drugs (multidrug resistance, MDR), exhibiting a higher migration capacity compared to that of MG63 cells. In MG63/VCR, LIMK1 is overexpressed at both the mRNA and protein levels, with a subsequent elevated cofilin phosphorylation. The knockdown of LIMK1 inhibited the migration of MG63/VCR cells, suggesting that LIMK1 dysregulation contributes to the invasion and metastasis potential of drug-resistant osteosarcoma [61]. MDR-associated genes (MDR1, MRP1, and BCL2) are upregulated in MG63/VCR. MDR1 expression was positively correlated with LIMK1 expression, as was LIMK1 silencing with an enhanced apoptosis in the MG63 cells treated with VCR. These data suggest that LIMK1 is implicated in the MDR of osteosarcoma, probably by inducing MDR1 expression and/or activation and by limiting apoptosis [126,133].

As Rac1/PAK1/LIMK1/cofilin and RhoA/ROCK/LIMK2/cofilin signalling pathways have been shown to be activated in OS pathology, these proteins, especially LIMKs, appear as promising therapeutic targets to treat this disease. Several inhibitors of these pathways have shown promising results. Sea cucumber polysaccharide fucoidan (Cf-Fuc) significantly diminished the migration and adhesion capacity of U2OS cells and the remodelling of the actin cytoskeleton. Cf-Fuc inhibited the phosphorylation of focal adhesion kinase (FAK) and paxillin, as well as LIMK1 and cofilin phosphorylation, and Rac1 activation [134]. 6-hydroxythiobinupharidine that was isolated from Nuphar pumilum also suppressed the migration of murine LM8 osteosarcoma cells by decreasing the expression of LIMK1 and cofilin phosphorylation [135]. Taken together, these results indicate a high potential for therapy-targeting LIM kinases.

#### 6.1.5. Glioblastoma

Glioblastomas (GBM) are the most common and aggressive primary brain tumour in adults, exhibiting a very high infiltration and a very poor prognosis, since their survival time is less than two years upon diagnosis. GBM derive from multiple cell types with neural stem-cell-like properties and require multi-modal therapies, including chemotherapy, radiotherapy, and surgical intervention, which are commonly used [136]. However, despite this tough treatment, most GBM recur; hence, it is necessary to understand the molecular mechanisms of the pathology and to develop new and specific therapies for GBM.

Several studies have shown that LIMK1 and LIMK2 are overexpressed and overactivated in patient tissue and different GBM cell lines. Park et al. performed a microarray analysis of the normal brain versus mesenchymal GBM and observed an increase in LIMK and cofilin expression [137]. An analysis of the gene expression data from TCGA and REMBRANDT, and a clinically annotated dataset from several groups confirmed these data: the clinical features of GBM are correlated with changes in LIMK expression. LIMKs are upregulated at the mRNA and protein levels in GBM, and patients with gliomas that exhibit downregulated LIMKs have a better overall survival [60,137,138]. The upregulation of LIMKs is also observed in different GBM cell lines (U87, T98G and U118) [137,138].

As they have been shown to be upregulated in GBM cell lines and patients, and as they play important roles in cell polarisation, migration, and invasion, LIM kinases were thought to be suitable therapeutic targets for the treatment of GBM by inhibition. The knockdown of both the LIM kinases with shRNA significantly reduced the invasion of the GBM cell lines and of the human GBM tumour-initiating cells (TICs), which is a more clinically proximal culture model, as it forms infiltrative tumours when xenografted in mice. Moreover, the tumours derived from LIMK1/LIMK2-knockdowned TICs were significantly smaller, as they grew slower compared to the tumours derived from the control TICs. They also spread at lower rates (with a reduced invasion propensity), resulting in an increase in survival [60]. However, surprisingly, Erktulu et al.’s RNA analysis of the paraffinized tumour tissue from 98 patients that were diagnosed with GBM pointed out that LIMK1 mRNA upregulation is correlated with an increased survival (more than six months) upon surgery [139].

Several proteins have been shown to regulate the LIMK and cofilin in GBM: PKCζ and intersectin-1 [116,140]. PKCζ is an atypical PKC. An interaction between endogenous PKCζ and LIMK1, but not LIMK2, was shown in the LN229 glioblastoma cell line upon EGF stimulation. The knockdown of PKCζ by siRNA in cell lines, as well as in mouse xenografted tumours, resulted in the specific and important impairment of glioblastoma cell migration and invasion. PKCζ silencing impaired the phosphorylation of LIMK and cofilin, indicating that PKCζ regulates both the cytoskeleton rearrangement and cell adhesion, thereby contributing to cell migration via the LIMK-cofilin pathway [116]. Intersectin-1 coordinates the endocytic trafficking with the actin assembly machinery. Like PKCζ, the knockdown of intersectin-1 by siRNA in GBM cell lines leads to a decrease in PAK, LIMK, and cofilin phosphorylation, as well as in migration and invasion. Some partners of LIMKs, PTEN, and Nf1 could also trigger the implication of LIMKs in GBM [140]. A loss of PTEN function is associated with a poor survival in anaplastic astrocytoma and glioblastoma. An overexpression of the epidermal growth factor receptor (EGFR) in heterozygous PTEN KO mice leads to the development of invasive glioma, which is very similar to human glioblastoma, indicating that PTEN plays a role in glioma progression [141]. Since it has been shown in prostate cancer cell lines that LIMK2 and PTEN are engaged in a negative regulatory loop, where PTEN promotes the degradation of LIMK2 by ubiquitination and LIMK2 inhibits PTEN function and promotes its degradation, an implication of LIMK2 in PTEN-loss-driven GBM aggressiveness might be considered [109]. Neurofibromin 1 (Nf1) is one of the most mutated genes in GBM. An interaction between Nf1 and LIMK2 that results in the inhibition of LIMK2 with Nf1 has been shown [142]. Similarly, Nf1 was shown to inhibit the PAK4/LIMK1/cofilin pathway [143]. The Nf1 mutations encountered in GBM may result in a loss of its inhibitory activity on LIMKs, also contributing to the overactivation of LIMKs observed in GBM.

Several chemical inhibitors of LIMKs were also used in GBM. Park et al. used LIMKi 3 (BMS-5) on two GBM cell lines and they showed significant decreases in the viability of the GBM cells treated, with no cytotoxic effects on the normal astrocytes [137]. BMS-5 increased the adhesion of GBM cells, and decreased their migration and invasion, as well as their cofilin phosphorylation [137]. Schulze et al. also used LIMKi 3 (BMS5) and observed different responses depending on the cell lines (chemo-sensitive NCH644 vs. chemo-resistant NCH421k) [144]. Cucurbitacin I, which inhibits cofilin phosphorylation through an unknown mechanism, and alantolactone, which inhibits cofilin in a dose-dependent manner, were also tested on different GBM cell lines, and similar effects were observed: a decrease in migration and invasion, and an increase in apoptosis and adhesion [137,145]. These results are in favour of the regulation of the GBM invasive motility and tumour progression of LIM kinases.

### 6.2. Neuronal Diseases

Several studies on knockout animal models have made clear the implication of LIM kinases, and more particularly LIMK1, in neurodevelopment and synaptic plasticity. Integrating multiple pathways to regulate finely dendritic spines, synaptic plasticity [84], learning, memory [146], and neuron migration [147] through actin dynamics, LIM kinase dysregulations have been reported in many brain diseases.

#### 6.2.1. Alzheimer’s Disease

Alzheimer’s disease (AD) is a neurodegenerative disorder and the most common form of dementia, accounting for 60 to 80 percent of cases [148]. It is characterised by a progressive loss of cognitive functions, such as memory and visuo-spatial impairments [148,149], accompanied by neuropsychiatric symptoms including depression, apathy, and anxiety [148,150,151]. AD is typically hallmarked by early synaptic loss, an accumulation of the extracellular deposits of beta-amyloid peptides (Aβ_42_) in senile plaques, and the development of neurofibrillary tangles composed of hyperphosphorylated tau [152,153,154]. Aβ_42_ results from the proteolysis of amyloid precursor proteins (APP) with secretases and their abnormal production and aggregation is directly linked to neurotoxicity, driving the synaptic damages and dendritic loss characteristic of AD [155,156]. The formation of aberrant focal adhesion structures, cofilin–actin rods [157], Hirano bodies composed of ADF/cofilin and actin [158,159], and abnormal actin stabilisation suggest that actin cytoskeleton remodelling has a role in pathological neuroplasticity [160,161].

Synaptic strength and activity are deeply linked to spine morphology [87]. Several studies have shown an increase in phospho-LIMKs and phospho-cofilin in the AD brain area of patients or AD animal models by using western blot or immunofluorescence [155], more particularly in the post-synaptic compartment of excitatory synapses [161], as well as an increase in the ROCK protein level [162,163]. As the antibodies against phospho-LIMKs recognise both the phospho-Thr508 of LIMK1 and the phospho-Thr505 of LIMK2, it is not possible to discriminate the role of the activation of each protein. Both are probably concerned. These studies also pointed out an aberrant remodelling of the actin filaments (with an increased stabilisation) within spines, leading to neuritic dystrophy, with a reduction in the neuritic network and an impairment of the synaptic strength and plasticity of rat hippocampal neuron structures [155]. On the other hand, the expression of LIMK1 in hippocampal excitatory neurons increases cofilin phosphorylation, rescues the impairments of long-term potentiation, and improves the social memory of 3-month-old APP/PS1 transgenic mouse models [164].

Fibrillar Aβ (fAβ) treatment also activates Cdc42/Rac1/PAK1 signalling pathway, linking fAβ to actin dynamics via a different pathway to the Rho/ROCK pathway. Mendoza-Naranjo et al. established a time course activation of Cdc42/Rac1/PAK1 upon Aβ treatment [156]. LIMK1 was activated (phosphorylated) concomitantly with PAK1. An activation of Cdk5 was also depicted, but later in the process, leading to hyperphosphorylation and the inactivation of PAK1. Surprisingly, no change in the cofilin phosphorylation level was observed at any time. The authors suggest an implication of the slingshot phosphatase SSH1 in this process and point out a fine balance between the kinase and phosphatase activation regulating cofilin phosphorylation, and subsequently actin remodelling upon Aβ treatment [156].

Moreover, the treatment of rat hippocampal neurons with fibrillar Aβ (fAβ) increased the level of inactivated cofilin (Ser-3 phosphorylated) and activated LIMK1 (Thr-508 phosphorylated) with abnormal actin remodelling, neuritic dystrophy, and cell death. The inhibition of cofilin phosphorylation with S3 peptide, a specific competitor of Ser3 cofilin phosphorylation, diminished the effect of fAβ on actin filament remodelling and neuronal degeneration, indicating that LIMK1 plays a role in fAβ-induced neurotoxicity [155,161,165].

The pharmacological inhibition of ROCK by fasudil counterbalances Aβ-induced features, especially synaptic loss [161]. Henderson et al. showed that ROCK2 is responsible for spine loss via LIMK1 activation, while ROCK1 proceeds via the myosin actin pathway [163]. Fasudil was also used on an AD rat model induced by streptozoticin treatment. Fasudil reverses learning and memory deficits, synaptic structure degeneration, and phospho-LIMK2 and phospho-cofilin increases [146]. The pharmacological inhibition of LIMK1 by SR7826 [166] on a hAPP mouse model of AD protects against Aβ-induced neuronal hyperexcitability, Aβ-induced spine degeneration, and rescued hippocampal thin spine loss. Synaptosome fractions of hippocampal tissue homogenates from a hAPP mouse treated with SR7826 showed reduced levels of phospho-cofilin. Interestingly, the ROCK knockdown by the siRNA in neurons or the ROCK heterozygous knockout mice suppressed the endogenous production of Aβ by enhancing APP lysosomal degradation [162,163].

The nuclear receptor Nurr1 was shown to be associated with AD [167]. Indeed, upon the Nurr1 knockdown by shRNA in 5xFAD, an AD mouse model, the features of AD pathology become worse, whereas the overexpression of Nurr1 (via lentiviral vector) or its pharmacological activation (amodiaquine agonist) reduce these features. LIMK1 has been shown to interact with Nurr1 by co-immunoprecipitation experiments, resulting in the inhibition of the Nurr1 transcriptional activity [41]. Further investigations on the LIMK1/Nurr1 relationship within an AD context could bring new insights into this field.

#### 6.2.2. Parkinson’s Disease

Parkinson’s disease (PD) is the second most common neurodegenerative disease after Alzheimer’s disease, and the most common movement disorder [81,168]. It is characterised by bradykinesia (slowed movements), resting tremor, rigidity, and postural instability [169]. These impairments are mainly due to the loss of dopaminergic neurons in the substantial nigra (SN) pars compacta, which can be imputed to the development of abnormal protein aggregates called “Lewy bodies” (LBs) and degenerated neurites (“Lewy neurites”, or LNs) [170]. LBs are mainly composed of misfolded alpha-synuclein (α-synuclein), an abundant protein of the central nervous system, which is highly enriched in pre-synaptic nerve terminals [171,172].

As actin plays a crucial role in synaptic function, a link between actin and alpha-synuclein was investigated [173]. A direct interaction between cofilin and alpha-synuclein was detected through the co-immunoprecipitation of rat brain homogenates [174].

In a cell-free actin polymerisation in vitro assay, alpha-synuclein was shown to decrease actin polymerisation and to increase actin depolymerisation by sequestering actin monomers. Furthermore, Yan et al. showed that a cofilin/alpha-synuclein interaction promotes its aggregation [175]. These mixed fibrils that associate alpha-synuclein and cofilin are more compact and more potent in seeding alpha-synuclein aggregation, as their uptake is increased and more deleterious effects are observed on neuron morphology (especially neurites).

Alpha-synuclein is also released from neurons, and this extracellular alpha-synuclein was shown to activate the Rac/PAK2/LIMK/cofilin pathway via its interaction with GRP78, a chaperone heat shock protein [176]. Indeed, when hippocampal neurons are incubated with recombinant alpha-synuclein, an increase in phospho-PAK2 and phospho-cofilin was observed, along with an accumulation of lamellipodia-like actin protrusions along the neurites and at their tips. These effects were lost when the GRP78 was downregulated.

The Parkin gene is mutated in autosomal recessive juvenile Parkinsonism (ARJP) and early-onset Parkinsonism. The Parkin protein acts as an E3 ubiquitin ligase. Lim et al. [177] have shown that Parkin interacts with LIMK1 in HEK293 transfected cells, with the C-terminal part of Parkin and the N-terminal part of LIMK1 (LIM-LIM-PDZ) being involved in this interaction. Even though they focus on LIMK1, they showed a similar interaction between LIMK2 and Parkin. The authors could not detect an endogenous interaction on the mouse brain lysates. Parkin ubiquitinates LIMK1 and reduces the LIMK1-induced cofilin phosphorylation in dopaminergic neuronal BE(2)-M17, but not in HEK293 cells. Parkin also reduces the LIMK1-induced actin stress fibres in COS7 cells. Although LIMK1 does not phosphorylate Parkin, it lowers its ubiquitin ligase activity on itself and on p38-MAPK, one of its substrates. Therefore, Parkin and LIMK1 regulate each other.

Nurr1 is highly expressed in dopaminergic neurons, where it plays a role in cell differentiation and survival. Defects in Nurr1 expression have been associated with PD in animal models and in the brain samples from PD patients [178,179,180]. As mentioned earlier, LIMK1 interacts with Nurr1 and inhibits its activity upon phosphorylation [41]. Thus, the inhibition of the LIMK1-driven repression of Nurr1 activity could represent an opportunity for PD therapy.

#### 6.2.3. Autism Spectrum Disorders

Autism spectrum disorders (ASD) is a term used to designate a multitude of early-appearing behaviour impairments regarding social communication. These disorders are also characterised by repetitive and unusual sensory–motor behaviours, even though individuals with ASD display very different features to one to another [181]. Fragile X syndrome (FXS) is the most common inherited form of intellectual disability and autism spectrum disorder. Patients with FXS display severe behavioural dysfunctions, a hypersensibility to sensory stimuli, anxiety, poor language development, and epileptic seizures. FXS is caused by the silencing of the FMR1 gene, which encodes the fragile X messenger ribonucleoprotein 1 protein (FMRP). FMRP has a central role in gene expression, through the regulation of the translation of mRNAs, which are suspected to be involved in the development and maintenance of neuronal synaptic connections [182,183]. Dysregulation of the synaptic actin cytoskeleton, dendritic spines morphology, and synaptic plasticity is particularly described in FXS subjects [184,185,186].

Mutations in the RAC1 gene were reported in ASD patients and several studies have shown that the disruption of Rac1 signalling in animal models leads to ASD-like behaviours [187]. Moreover, Rac1 levels and activity were shown to be significantly higher in FXS patients [188]. Pyronneau et al. showed that the Rac1/Pak1/LIMK/Cofilin pathway is implicated in the aberrant neuronal structures of patients with FXS [189]. Indeed, increased Rac1/PAK1/LIMK/Cofilin signalling in the somatosensory cortex of FMR1 KO mice has been linked to aberrant spine morphology and density. Notably, PAK1 inhibition rescued cofilin regulation, glutamatergic signalling, and sensory processing [189,190,191], which led to thinking that a deficiency in the PAK signalling pathway might play a role in human FXS pathogenesis. Moreover, PAK2^+/−^ mice display decreased synapse densities, defective long-term potentiation, and autism-related behaviours, and PAK2 nonsense mutations and deletions that impaired the PAK2 function have been found in large cohorts of patients with ASD [86].

BMPR2 mRNA is a target of FMRP [192]. BMPR2 is known to bind and activate LIMK1 in a pathway that stimulates actin reorganisation to promote neurite outgrowth and synapse formation [193]. The depletion of FMRP increased the BMPR2 abundance, which was observed in *Drosophila* and mouse models of FXS. The morphological defects associated with BMPR2-LIMK1 signalling [192,194] were rescued by BMPR2 heterozygosity or LIMK1 inhibition. A BMPR2 increase was also found in the prefrontal cortex of FXS patients [192], suggesting that LIMK1 could play a preponderant role in the actin-driven anomalies within the neuronal development of FXS patients.

Conversely, Yao et al. showed a downregulation of LIMK1 in the plasma samples of ASD patients [195], which happened to be contrary to the observations made regarding the BMPR2 increase in FXS. This difference could be the result of a different cellular context, since LIMK1 is mostly expressed in the central nervous system. The deregulation of LIMK1 expression might then be tissue-specific.

#### 6.2.4. Schizophrenia

Schizophrenia (SZ) is a common, severe psychiatric disorder characterised by core features such as delusions and hallucinations, and by behavioural impairments including social withdrawal and depressive moods. SZ is a multifactorial disease caused by genetic and/or environmental factors [196]. A disturbance in glutamatergic functions and the deregulation of the actin cytoskeleton, especially regarding neuronal and synaptic dysfunctions, has been well characterised within SZ [197,198].

In the dorso lateral prefrontal cortex (DLPFC) of patients with SZ, the CDC42 gene is notably underexpressed [199,200,201]. Monkey models that were chronically exposed to antipsychotic medications showed no alteration in their CDC42 expression levels [199]. An analysis of CDC42-related gene expression displayed increased mRNA levels of LIMK1 and LIMK2 in the DLPFC of subjects with schizophrenia, implying that the CDC42/PAK/LIMK pathway has a role in the spine deficits in the DLPFC of SZ subjects [201]. Accordingly, another study showed an overexpression of LIMK2 in the nucleus accumbens, prefrontal cortex, and hippocampus of a neonatal ventral-hippocampal lesion rat model of SZ [202,203].

The neuronal inhibition of the 14-3-3 protein in KO mice leads to behavioural defects that correspond to the core symptoms of SZ, which could be imputed to an alteration of the actin dynamics. Indeed, following this inhibition, the mice showed a reduction in dendritic complexity and spine density, as well as downregulated levels of phosphorylated cofilin [204]. Indeed, Gohla and Bokoch showed that the isoform 14-3-3ζ was able to bind phosphorylated cofilin at Ser3 and protect it from phosphatase-mediated dephosphorylation [29]. 14-3-3ζ was also shown to interact directly with LIMK1, which could prevent its kinase activity on cofilin [28] (Figure 8).

LIMK1 seems to play another role in SZ [205] notably through its interaction with neuregulin 1 (NRG1), a protein involved in synaptic plasticity and in the expression/activation of neurotransmitter receptors, including glutamate receptors, which exerts its synaptic activity through LIMK1/cofilin-mediated actin reorganisation [206]. NRG1 mutations have been linked to SZ [207,208]. NRG1 transgenic mice, in which NRG1 is upregulated in the neurons of their forebrains, exhibited increased LIMK1 activity, reduced spine density, and glutamatergic impairment. The pharmacological inhibition of LIMK1 diminished these effects on spine density and ameliorated the glutamatergic impairments [208,209] (Figure 8).

Gory-Fauré et al. showed the dysregulation of the LIMK1 expression in MAP6 KO mice, a mouse model of SZ which display social withdrawal and anxiety-like features [210]. Treatment with the LIM kinase inhibitor Pyr1 rescued these behavioural impairments [210], making LIMK1 a potential target for the treatment of schizophrenia.

#### 6.2.5. Williams–Beuren Syndrome

Williams–Beuren syndrome (WBS) is a rare genetic disorder that is characterised by distinctive craniofacial features, congenital heart disease, and cognitive and behavioural dysfunctions that include intellectual disability and hypersociability. WBS is caused by a heterozygous deletion of approximately 1.5 Mb at the chromosome 7q11.23, which leads to the loss of one copy of 25–27 genes, including LIMK1. The loss of LIMK1 is believed to be responsible for some of the neurological aspects of WBS, notably the impaired visual–spatial cognition and long-term memory dysfunctions [211]. Indeed, Gregory et al. showed that the hemideletion in Williams syndrome and the LIMK1 sequence variation in the general population alter the functional connectivity of the intraparietal sulcus, a visual processing region, in similar ways [212]. Moreover, LIMK1 KO mice displayed impairments in fear processing, long-term memory, and spatial learning [84,213], which are features that could be associated with WBS. Further studies would be necessary to determine the molecular pathways that lead to these cognitive impairments induced by the LIMK1 deletion in WBS.

#### 6.2.6. Amyotrophic Lateral Sclerosis

Amyotrophic lateral sclerosis (ALS) is a neurodegenerative disease characterised by progressive loss of motor neurons in the brain and spinal cord, leading to the paralysis of most of the body muscles. Death usually occurs upon respiratory paralysis, within 3 to 5 years after diagnosis [214]. ALS etiology is complex, with genetic and environmental components, but some genes, such as SOD1 and C9ORF72, have been markedly associated with ALS when dysregulated [214,215]. There is actually no efficient treatment for this disease [214], but a possible involvement of the dysregulation of the cytoskeleton in ALS pathogenesis opens up the way to new therapies [216].

First of all, C9ORF72 has been shown to interact with cofilin. And, C9ORF72-depleted cells and post-mortem brain samples from ALS patients exhibit enhanced cofilin phosphorylation. This increase results from the activation of the Rac1/PAK/LIMK/Cofilin pathway, since C9ORF72 modulates the activity of Rac1. In C9ORF72-depleted motor neurons, axonal actin dynamics are impaired, indicating that the Rac1/PAK/LIMK/Cofilin pathway has a role in C9ORF72-driven ALS [215] and that there is a therapeutic potential for LIMK inhibitors.

BMP-TGF-β signalling has also been associated with neuronal function, playing a role in synaptogenesis, axonal and dendritic growth, synaptic transmission, and neuronal survival. Disruptions in BMP/TGF-β signalling have been reported in a variety of neurological diseases, notably in Alzheimer’s disease, Parkinson’s disease, and amyotrophic lateral sclerosis (ALS) [217]. BMPR2, a BMP receptor, is known to bind and activate LIMK1 in a pathway that stimulates actin reorganisation, in order to promote neurite outgrowth and synapse formation [193]. The upregulation of BMP-TGF-β signalling could thus lead to an impairment of actin dynamics, which could result in defective axon and dendrite functioning.

Cdk5 is dysregulated in various neurological disorders, including Alzheimer’s disease, Parkinson’s disease, and amyotrophic lateral sclerosis (ALS) [218]. When it is upregulated in AD and ALS, Cdk5 causes tau and neurofilament protein hyperphosphorylation, leading to neuronal cell death [219]. The Cdk5 inhibition of PAK1 has been shown to decrease the LIMK1 activity and to increase the active cofilin pool, leading to abnormal actin remodelling. Indeed, there is some evidence regarding the involvement of actin regulation as a causative and dysregulated process in ALS [216,218].

Myosin-binding protein H (MyBP-H) is a component of the thick filaments of the skeletal muscle that has strong affinity for myosin. It has been shown to be upregulated in ALS. A high MyBP-H expression level was also associated with the abnormal expression of Rho kinase 2 (ROCK2), LIM domain kinase 1 (LIMK1), and Cofilin2, indicating that the ROCK2/LIMK1/Cofilin2 pathway might play a role in ALS pathogenesis in muscles [220].

### 6.3. Neurofibromatosis

Neurofibromatosis are genetic orphan diseases without any connection to each other, except for the development of tumours in the nervous system. To date, three neurofibromatosis have been described: neurofibromatosis type 1 (NF1, or von Recklinghausen disease), neurofibromatosis type 2 (NF2), and schwannomatosis (NF3). An LIMK implication has been reported in NF1 and NF2 and will be discussed here.

#### 6.3.1. Neurofibromatosis Type 1

Neurofibromatosis type 1 (NF1) is an autosomal-dominant disease and the most frequently diagnosed cancer predisposition disorder that involves the nervous system [221,222]. It is caused by inherited or de novo mutations in the gene encoding neurofibromin 1 (Nf1) [223], a GTPase-activating protein, which is highly expressed in the neuronal cells and acts a tumour suppressor by negatively regulating Ras pathways [224,225]. Neurofibromatosis type 1 is hallmarked by the development of tumours in the central or peripheral nervous system, with high risk of malignancy. It is also characterised by cognitive dysfunctions, including learning impairments, attention deficits, and dysfunctional social behaviours [222,225].

The GAP-related domain (GRD) of Nf1 is the effector of the downregulation of Ras pathways, which are involved in cell growth. The pre-GAP domain was reported to play a role in cytoskeleton remodelling. Indeed, the N-terminal extremity of Nf1 affects cell adhesion and migration by negatively regulating the Rac1/Pak1/LIMK1/cofilin pathway [143]. Nf1 also plays a role in the Rho/ROCK/LIMK2/cofilin pathway, acting as a negative regulator of this pathway. Its deletion has indeed been shown to activate the signalling cascade, affecting the actin cytoskeleton [226]. The SecPH domain of Nf1 interacts with LIMK2, but not LIMK1, inhibiting its phosphorylation and activation via ROCK [142]. In Nf1^−/−^ mouse embryonic fibroblasts (MEFs), an increase in phospho-LIMK and phospho-cofilin was shown [143] (Figure 9). Nf1 seems to play a role in the fine regulation of the actin cytoskeleton by regulating, in parallel, two signalling pathways that lead to ADF/cofilin phosphorylation. These pathways, when dysregulated, are known to be involved in various neuronal and synaptic pathological mechanisms, as well as in the progression of cancers. Taken together, these results could explain the pathological features associated with the disease, and LIMKs, as downstream effectors of these pathways, could be interesting therapeutic targets for NF1.

#### 6.3.2. Neurofibromatosis Type 2

Neurofibromatosis type 2 is an autosomal-dominant neoplasia disorder caused by mutations in the neurofibromatosis 2 (NF2) gene, encoding a tumour suppressor protein called Merlin. It is characterised by nervous system tumours, notably multiple schwannomas, especially in the vestibulocochlear nerve, meningioma, ependymomas, neurofibromas, peripheral neuropathy, ophthalmological lesion skin tumours, and hearing loss [227]. Merlin interacts with membrane-associated proteins and transmembrane receptors, where it regulates the formation of membrane domains and acts as a tumour suppressor by modulating signalling, as a scaffold protein, from the transmembrane receptors to the intracellular effectors, controlling cell proliferation and survival [228,229].

A Merlin loss of function is associated with the increased activity of Rac and p21-activated kinases (PAK), and the dysregulation of cytoskeletal organisation [230]. In mouse Schwann cells (MSC), in which *NF2* exon2 is deleted (*NF2^Δ^^Ex^^2^*) and Merlin function is lost, the levels of phosphorylated LIMK1, phosphorylated LIMK2, and the subsequent phosphorylated cofilin were upregulated. However, no direct interaction between Merlin and LIMKs has been shown so far. The reintroduction of wild-type *NF2* into these MSCs reduced the LIMK1 and LIMK2 levels. A reduction in the LIMK activity and/or protein levels decreases the *NF2*-deficient MSC viability, and the pharmacological inhibition of LIMKs by BMS5 decreases the viability of *NF2^Δ^^Ex^^2^* MSC, blocks the cell cycle progression in the G2/M phase, and decreases AURKA activation [231]. Taken together, these results suggest that LIM kinases play a role in Merlin-loss-driven cytoskeletal dysfunctions and could act as potential therapeutic targets for the treatment of NF2.

### 6.4. Viral Infections

The three main steps of a viral infection are: (i) its entry into the host cell, (ii) its replication within this cell, and (iii) the release of many virions/viral particles. Viruses need the help of the host cell cytoskeleton to be able to carry out their replication cycle. Thus, they have evolved to hijack the cytoskeletal network and use it to their advantage [232,233]. The LIM kinase implication in the context of viral infection is quite new, but there is evidence that they play a crucial role in the viral life cycle and could be of a therapeutic interest for certain viral infections [234].

The human immunodeficiency virus type 1 (HIV-1) has been widely studied over the last years, in particular has its implication in the fine regulation of actin cytoskeleton dynamics during each step of its replication cycle. The actin cytoskeleton has been reported to be involved in viral entry, reverse transcription, nuclear migration, the shuttling of viral components to the membrane, assembly, budding, and cell–cell transfer. Moreover, HIV has developed strategies to spatiotemporally lever the actin cytoskeleton network by upregulating, inhibiting, changing gene expression, cellular localisation, and even modulating the function of certain effectors, such as cofilin and LIMK1 [234,235].

During the course of the HIV-1 infection, cofilin is either inactivated or activated, depending on the step of the viral cycle. During early infection, the HIV-1 envelope glycoprotein gp120 interacts with the CD4 receptor and CXCR4 (or CCR5) co-receptor, activating RhoA and the Rac-dependent pathway, which leads to LIM kinase activation and cofilin inactivation [236]. Vorster et al. showed that, upon the HIV infection of resting CD4 T cells, a transient phosphorylation and the subsequent activation of LIMK1 is observed within 1 min, followed by a deactivation at 5 min, and a reactivation at 10 min [237]. Moreover, they showed that gp120 alone was sufficient to trigger this activation. The same features were observed in active CD4 T cells and macrophages. This activation of LIMK1 was shown to be mediated by Rac, PAK1, and PAK2, but not by PAK4, RhoA, or Cdc42. When LIMK1 was knockdown in the transformed and primary CD4 T cells, as well as in the stable human CD4 T cell line, CEM-SS, increases in the CXCR4 receptor at the membrane, endocytosis, and exocytosis were observed. Furthermore, the stable knockdown of LIMK1 by shRNA in CD4 T cells renders the T cells resilient against HIV infection [237]. Thus, LIMK1 activation leads to the inhibition of cofilin actin severing activity; hence, for early cortical actin skeleton polymerisation and CD4/CXCR4 receptor clustering, two mechanisms are required for virus entry into the host cell [236]. In the following steps of the HIV-1 infection, there is evidence of viral-driven cofilin dephosphorylation, leading to the depolymerisation of cortical actin, a mechanism by which the viral core penetrates and navigates through the host cell to the nucleus [236,238]. This mechanism seems to be responsible for the establishment of HIV-1 latency in resting CD4 T cells [239]. Indeed, Wu et al. showed an elevated level of activated cofilin in the resting CD4 T cells of the peripheral blood of patients infected with HIV-1 [240]. Yoder et al. showed that the incubation of T cells with S3 peptide, corresponding to the 16 first amino acids of human cofilin, leads to the decreased activity of LIMK1 on full cofilin, and an enhanced viral replication [238]. These data suggest that the viral replication is increased when the cofilin is less phosphorylated by the LIMK1 and consequently more active [238].

Nef, an actin-modifying HIV-1 protein, was shown to activate LIMK1 by increasing its phosphorylation on its Thr508, leading to an increase in cofilin phosphorylation and resulting in the inhibition of retinoid receptor-mediated reporter activity, which plays a crucial role in immune response [241].

The implication of the actin cytoskeleton in retroviral assembly and budding has also been established [242]. In order to identify new genes involved in HIV-1 virion assembly and release, Wen et al. performed a screening by using a siRNA library and measuring the HIV-1 particle release. LIMK1 and ROCK1 were identified in this screening. They confirmed that, when they depleted the LIMK1 in HeLa cells using siRNA, they observed a reduction in particle output. The same results were obtained with stable HeLa transfected with LIMK1 shRNA. The transfection of a shRNA-resistant FLAG-LIMK1 cDNA, restoring the LIMK1 protein levels, could reverse this effect, whereas the incubation with S3 peptide, a synthetic peptide which acts as a specific competitor for the ADF/cofilin phosphorylation by LIMK1, also leads to a decrease in particle output. ROCK1 is involved in this process, but not PAK1/2/4, as the HeLa cells treated with ROCK siRNA also exhibited a lower particle output. In the HeLa cells silenced for LIMK1, the HIV-1 virions accumulated at the plasma membrane. LIMK1 was shown to co-localise with the HIV-1 particle assembly sites and be incorporated within the HIV particles. Furthermore, the silencing of LIMK1 also resulted in a decrease in the HIV cell transmission from HeLa to the Jurkat cell lines [243]. Altogether, these results show that the ROCK/LIMK/cofilin pathway is involved in the HIV-1 particle release and the spread of the virus.

It is well documented that herpes viruses use cytoskeleton-regulating Rho GTPase signalling pathways during different phases of their replication cycle [244]. Moreover, herpes simplex virus 1 (HSV-1) seems to be able to trigger biphasic cofilin deactivation/activation in a way that is similar to HIV-1, to facilitate its entry into the host cell and its replication in neuronal cells, via RhoA and Cdc42 [245]. Indeed, the interaction between the virus envelope and host cell leads to EGFR/PI3K and Rho/ROCK pathway activation and, ultimately, to actin remodelling. The specific inhibition of these pathways significantly limits the virus infectivity [246]. Amentoflavone, a natural polyphenol compound found in many plants, which has a broad antiviral activity spectrum against several viruses, was tested on HSV-1. The treatment of neuronal cells with Amentoflavone induced a decrease in the phospho-cofilin, leading to an impaired reorganisation of the F-actin and viral infection. Amentoflavone also leads to a decrease in the transport of viral particles from the plasma membrane to the nucleus [247].

As all these studies have highlighted the direct involvement of LIMK1, via its activation, in the viral infection process, inhibitors of LIMK1 have been developed as a new approach to block viral infection. Yi et al. have designed a set of 25 new LIMK inhibitors, and 8 of them appeared to reduce HIV infection [248]. They further characterised their best lead, R10015, and showed that it blocks viral DNA synthesis, viral nuclear migration, and virion release. R10015 was shown to inhibit multiple viruses, including Zaire ebolavirus (EBOV), Rift Valley fever virus (RVFV), Venezuelan equine encephalitis virus (VEEV), and herpes simplex virus 1 (HSV-1), suggesting that the LIMK implication in the viral life cycle concerns many viruses [248]. Thus, LIMK inhibitors appear as a potential new class of broad-spectrum antiviral drugs.

Furthermore, there is growing evidence about the implication of the HSV-1 infection in the occurrence of neurological diseases such as Alzheimer’s disease. As reviewed by Wan et al., HSV-1 DNA co-localises with the senile plaques in AD brains, and some HSV proteins interact with Aβ and facilitate its production, as well as that of its precursor proteins [249]. Since cofilin dysregulation is a major factor in AD pathogenesis, as well as in virus infection, it could be the link between HSV-1 infection and AD development. Targeting cofilin or LIMKs could be a new strategy for treating patients infected with HSV-1 and exhibiting AD features.

### 6.5. Reproduction Troubles

LIMKs have been shown to play a role in the function of the male urogenital system, as well as in its associated defects. All of these features are well described in the review of Pak et al., which belongs to the Special Issue LIM Kinases: From Molecular to Pathological Features [250].

## 7. Conclusions

Because they are implicated in the dynamics of actin filaments and microtubules, LIM kinases play a crucial role in the remodelling of the cytoskeleton and, thus, in the physiology of the cell. Originally described as being located downstream of several signalling pathways involving members of the Rho family of GTPases, it appeared that LIMKs are, in fact, capable of integrating signals from multiple partners, some of them without an obvious link with their role in the cytoskeleton reorganisation. In fact, LIMKs are at the heart of a complex network of cell signalling pathways. The dysregulation of these interactions has linked them to severe pathologies such as cancer, neurological diseases, and viral infections, making LIMKs a node with strong therapeutic potential (Figure 10). These studies have paved the way for the development of small inhibitory molecules that are capable of modulating the activity of these kinases, but unfortunately, despite promising preliminary results with different mouse models, none have made it through clinical trials. It is now necessary to go further in the direction of understanding the molecular mechanisms that link LIM kinases to the pathogenic phenomenon, and to have a better understanding of their own functioning, in the hope of developing new, tailored, and innovative drugs that target LIM kinases.

## Figures and Tables

**Figure 1 cells-12-00805-f001:**
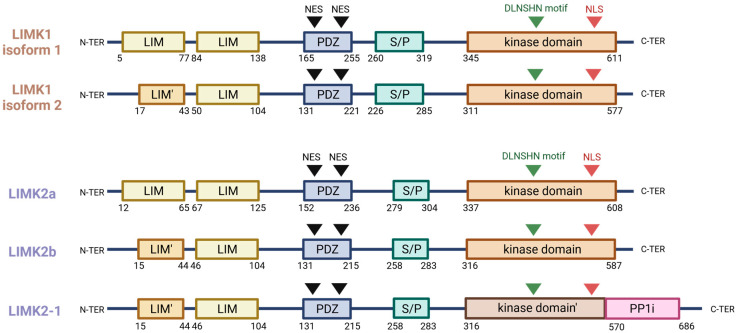
Schematic structures of human LIM kinases, LIMK1, and LIMK2, and their isoforms. Nuclear export signals (NES) are shown with a black arrowhead, DLNSHN motif is shown with a green arrowhead and nuclear localisation signal (NLS) is shown with a red arrowhead, ‘ stands for truncated domains.

**Figure 2 cells-12-00805-f002:**
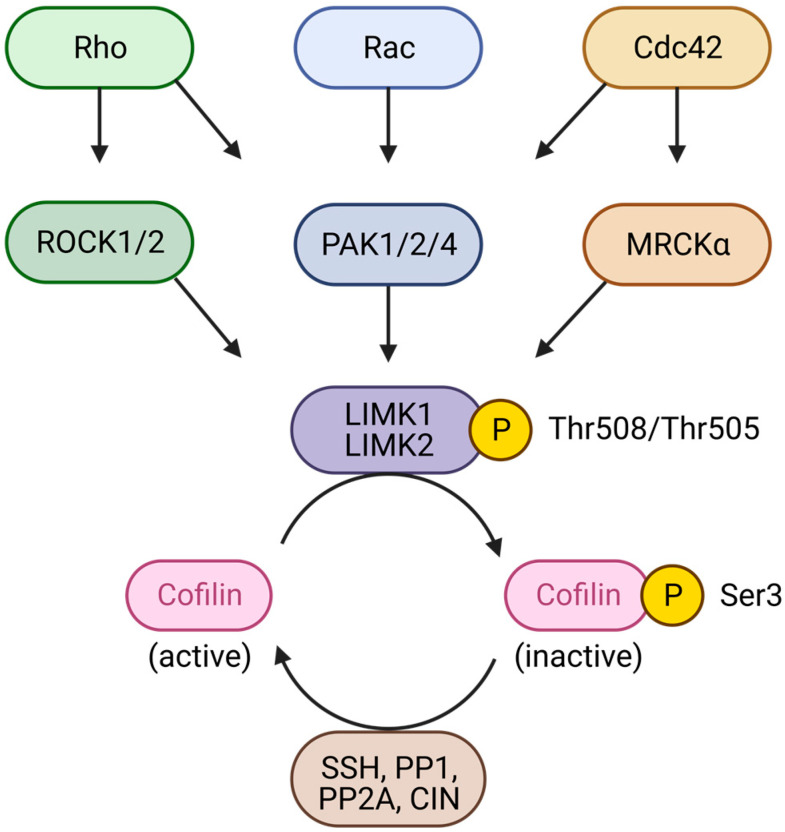
Cofilin regulation by LIM kinases, downstream of Rho GTPase family signalling pathways.

**Figure 3 cells-12-00805-f003:**
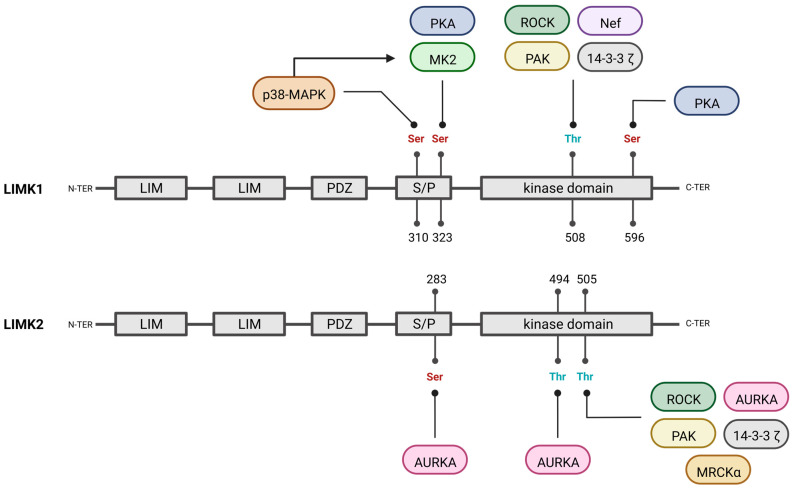
LIM kinase known phosphorylation sites and the kinases involved in these processes.

**Figure 4 cells-12-00805-f004:**
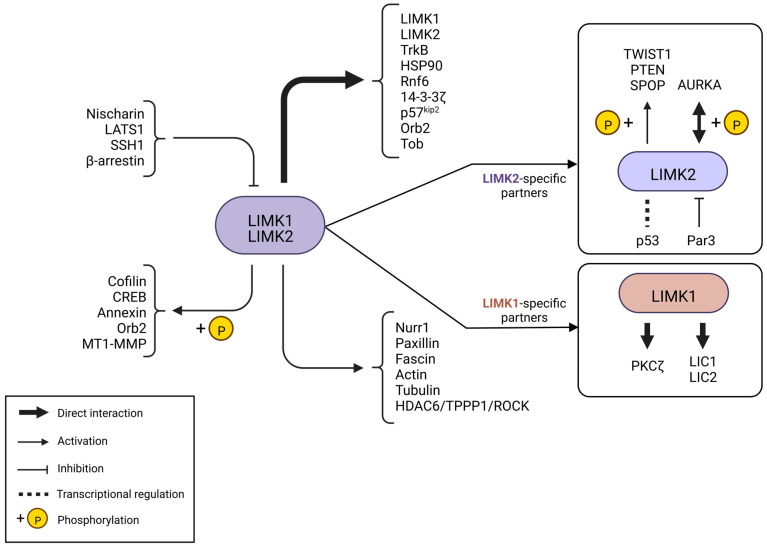
Partners of LIM kinases.

**Figure 5 cells-12-00805-f005:**
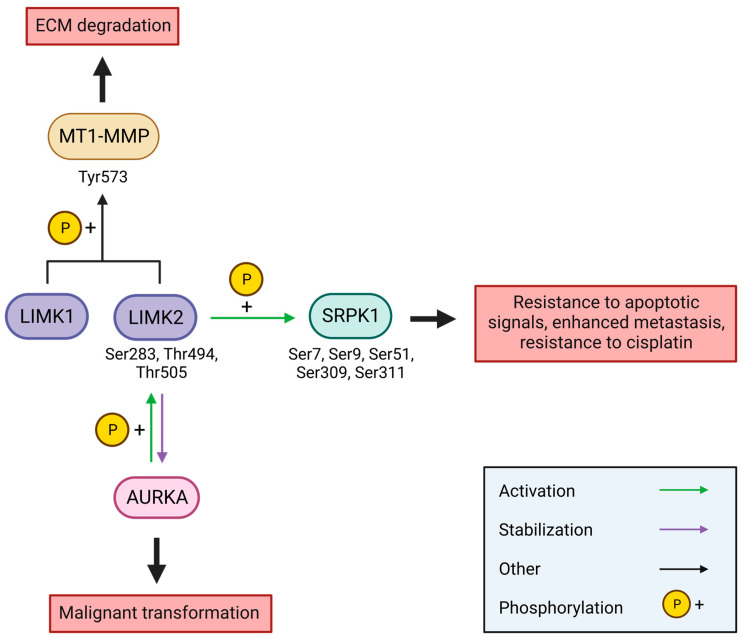
LIM kinases molecular implication in breast cancer. LIMK2 sites phosphorylated by AURKA are mentioned (Ser283, Thr494, and Thr505), as well as SRPK1 sites phosphorylated by LIMK2 (Ser7, Ser9, Ser51, Ser309, and Ser311), and MT1-MMP site phosphorylated by LIMK1 and LIMK2 (Tyr573).

**Figure 6 cells-12-00805-f006:**
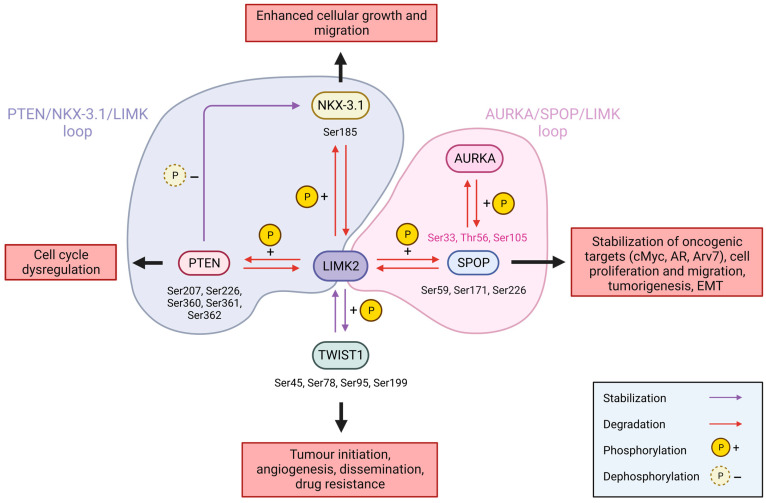
LIM2 molecular implication in prostate cancer. PTEN/KNX-3.1/LIMK and AURKA/SPOP/LIMK feedback loops are depicted with blue and pink areas, respectively. PTEN sites phosphorylated by LIMK2 (Ser207, Ser226, Ser360, Ser361, and Ser362), TWIST1 sites phosphorylated by LIMK2 (Ser45, Ser78, Ser95, and Ser199), as well as SPOP sites phosphorylated by LIMK2 (Ser59, Ser171, and Ser226), and NKX-3.1 site phosphorylated by LIMK2 (Ser185) are mentioned in black. SPOP sites phosphorylated by AURKA (Ser33, Thr56, and Ser105) are mentioned in pink.

**Figure 7 cells-12-00805-f007:**
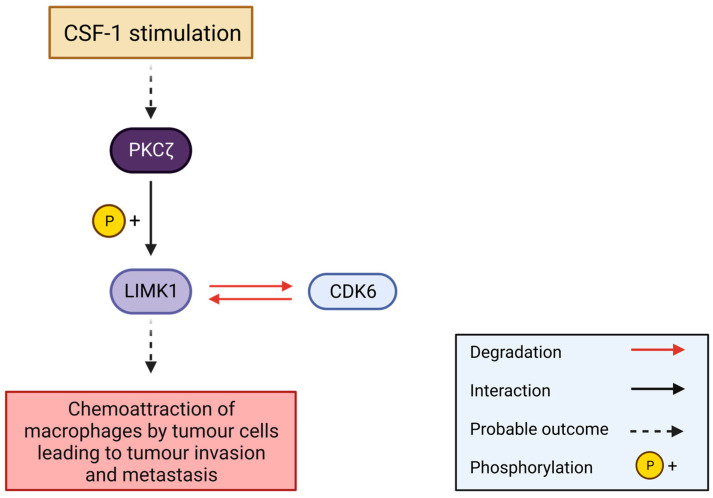
LIM kinases implication in the development of leukaemia.

**Figure 8 cells-12-00805-f008:**
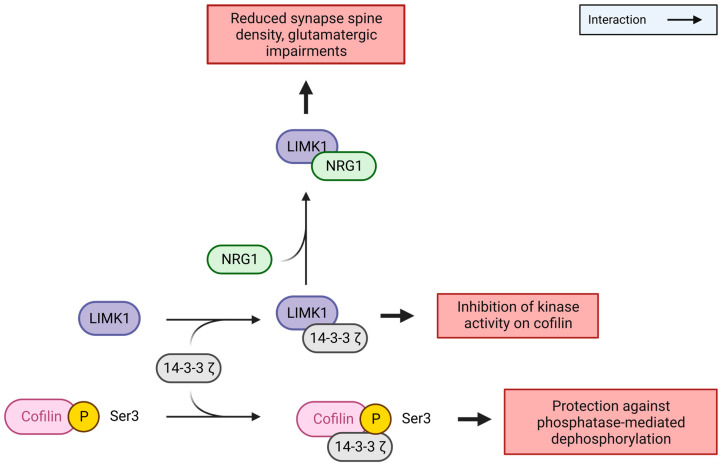
LIMK1 molecular implication in schizophrenia.

**Figure 9 cells-12-00805-f009:**
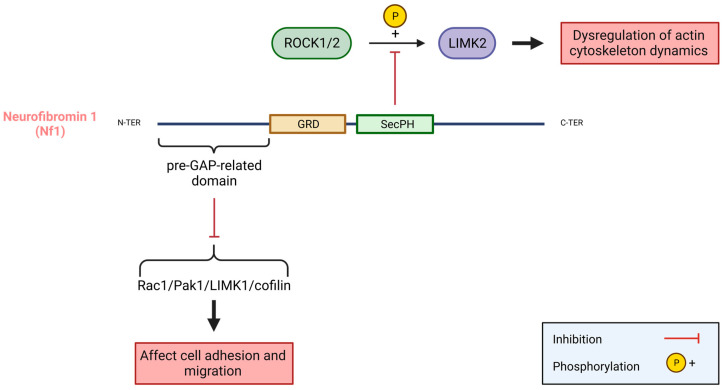
Interplay between Nf1 and LIM kinases, a key role in the development of neurofibromatosis type 1.

**Figure 10 cells-12-00805-f010:**
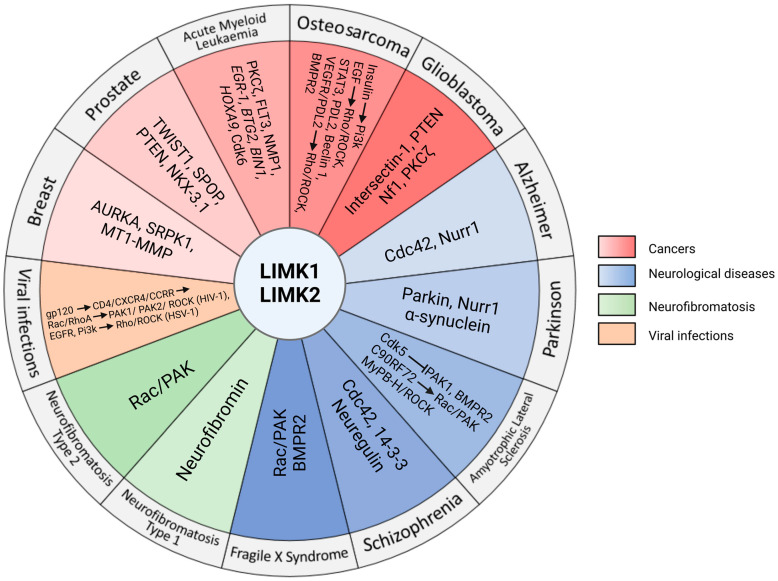
Involvement of LIM kinases in different pathologies. LIMK partners and LIMK mis-regulated signalling pathways implicated into these pathologies are also highlighted.

## Data Availability

Not applicable.

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
