# Peer review of "LIM Kinases, LIMK1 and LIMK2, Are Crucial Node Actors of the Cell Fate: Molecular to Pathological Features"

_cells, 2023, doi:10.3390/cells12050805_

Round 1
Reviewer 1 Report
This is a very well documented topical review, which I highly recommend for publication as soon as the following minor points are addressed.
English (language, grammar, spelling)
· P2L68 « Is missing » : lacks
· P4L51 « 14-3-3 proteins family » : 14-3-3 protein family
· P5L159: LIM kinase activity is finely up- but also down-regulated
· P7L229 “ a calcium‐ and phospho‐lipid‐binding proteins”
· P8L301:” During interphase, LIMK2 is diffuse overall the cytoplasm” : During interphase, LIMK2 is diffused throughout the cytoplasm.
· Localization, not localisation
· P9L328: “but in a lesser extent” : but to a lesser extent
· P6L174: “and none of the catalytic activities of these proteins are required for this interaction”. Not clear. Do the authors mean that the catalytic domain of LIMKs is required for the interaction? Or those of SSHs? Or both?
· P9L367: “The produced LIMK1 N‐ terminally truncated fragments are constitutively active and stimulates membrane blebbing when overexpressed, ultimately leading to cell death”
· P10L386 “This increase, as well as apoptosis induction, is lost when cells are pre‐treated with the ROCK inhibitor Y27632 [71].”
· P18L766 : “GBM derives from multiple cell types” GBM refers to glioblastomas, thus it should be written “ “GBM derives from multiple cell types”
· P18L805: “Overexpression of Epidermal growth factor receptor (EGFR) in heterozygous PTEN KO mice lead…” Leads (instead of lead)
· P21L918 “an increase in phospho-PAK2 and phospho-cofilin were observed” : an increase in phospho-PAK2 and phospho-cofilin was observed
Typo
· P7L224 (Figrue 4)
Figures
· Figure 2 : the size of the fonts is not homogeneous: for example, the font’s size of P-cofilin is smaller than the font size of cofilin
Other suggestions
· P2L71 “A testis‐specific LIMK2 isoform, tLIMK2, has also been described in mice [18].” The reference(18) is not exactly the good one, which is : Takahashi, H., Koshimizu, U., & Nakamura, T. (1998). A novel transcript encoding truncated LIM kinase 2 is specifically expressed in male germ cells undergoing meiosis. Biochemical and biophysical research communications, 249(1), 138–145. https://doi-org.insb.bib.cnrs.fr/10.1006/bbrc.1998.9094
· P2L71 In the context of this review, it may be informative that the authors specify how tLIMK2 is different from other forms. For example: A testis‐specific LIMK2 isoform, tLIMK2, that lacks LIM domains at the N-terminus, due to usage of a testis-specific alternative initiation exon, has also been described in mice
· A transition sentence is missing L152. For instance: An activation of LIMK2 by Aurora kinase A(AURKA) has been described (ref) . As AURKA is well known …
· P6L217 : “This trimeric complex leads to HDAC6 inhibition and consequently to microtubule stabilization” Microtubule acetylation is not responsible of their stabilization (See Janke, C., & Montagnac, G. (2017). Causes and Consequences of Microtubule Acetylation. Current biology : CB, 27(23), R1287–R1292. https://doi-org.insb.bib.cnrs.fr/10.1016/j.cub.2017.10.044)
· P8L307:” LIMK localisation was also studied during mitosis [61]. This transitional sentence is inappropriate because the previous paragraph already describes the localization of LIMK1 and LIMK2 during mitosis.
· P9L337 “Indeed, inhibition of LIMK1 activity during mitosis leads to a delay in mitotic progression (metaphase to anaphase) and irregular spindle positioning [65,66]” : also shown in Prudent et al, 2012
· P10L390-392: “Death‐associated protein kinase (DAPK), a protein implicated in programmed cell death, could act as a scaffold protein for the LIMK‐cofilin complex in TNF‐induced apoptosis, which is abrogated once cells are committed to apoptosis [72].” Not clear: what is abrogated?
· P15, leukaemia part : Reports of following publications are missing :
· Djamai, H., Berrou, J., Dupont, M., Kaci, A., Ehlert, J. E., Weber, H., Baruchel, A., Paublant, F., Prudent, R., Gardin, C., Dombret, H., & Braun, T. (2021). Synergy of FLT3 inhibitors and the small molecule inhibitor of LIM kinase1/2 CEL_Amide in FLT3-ITD mutated Acute Myeloblastic Leukemia (AML) cells. Leukemia research, 100, 106490. https://doi-org.insb.bib.cnrs.fr/10.1016/j.leukres.2020.106490
· Berrou, J., Dupont, M., Djamai, H., Adicéam, E., Parietti, V., Kaci, A., Clappier, E., Cayuela, J. M., Baruchel, A., Paublant, F., Prudent, R., Ghysdael, J., Gardin, C., Dombret, H., & Braun, T. (2022). Preclinical Evaluation of a Novel Small Molecule Inhibitor of LIM Kinases (LIMK) CEL_Amide in Philadelphia-Chromosome Positive (BCR::ABL+) Acute Lymphoblastic Leukemia (ALL). Journal of clinical medicine, 11(22), 6761. https://doi-org.insb.bib.cnrs.fr/10.3390/jcm11226761
Author Response
We thank the reviewers for their critical and careful reading and evaluation of our manuscript. Their comments improved and enriched the submitted manuscript, which we revised accordingly. Below we provide a detailed point-by-point response to each point raised in the reviewers' reports. The line numbers refer to the revised version of the manuscript with tracked changes.
Reviewer 1
This is a very well documented topical review, which I highly recommend for publication as soon as the following minor points are addressed.
English (language, grammar, spelling)
P2L68 « Is missing » : lacks => changed
P4L51 « 14-3-3 proteins family » : 14-3-3 protein family => changed
P5L159: LIM kinase activity is finely up- but also down-regulated => changed
P7L229 “ a calcium‐ and phospho‐lipid‐binding proteins” => changed
P8L301:” During interphase, LIMK2 is diffuse overall the cytoplasm” : During interphase, LIMK2 is diffused throughout the cytoplasm. => changed
Localization, not localization => changed
P9L328: “but in a lesser extent” : but to a lesser extent => changed
P6L174: “and none of the catalytic activities of these proteins are required for this interaction”. Not clear. Do the authors mean that the catalytic domain of LIMKs is required for the interaction? Or those of SSHs? Or both? => changed
P9L367: “The produced LIMK1 N‐ terminally truncated fragments are constitutively active and stimulates membrane blebbing when overexpressed, ultimately leading to cell death” => changed
P10L386 “This increase, as well as apoptosis induction, is lost when cells are pre‐treated with the ROCK inhibitor Y27632 [71].” => changed
P18L766 : “GBM derives from multiple cell types” GBM refers to glioblastomas, thus it should be written “ “GBM derives from multiple cell types” => changed
P18L805: “Overexpression of Epidermal growth factor receptor (EGFR) in heterozygous PTEN KO mice lead…” Leads (instead of lead) => changed
P21L918 “an increase in phospho-PAK2 and phospho-cofilin were observed” : an increase in phospho-PAK2 and phospho-cofilin was observed => changed
Typo
P7L224 (Figrue 4) => changed
Figures
Figure 2 : the size of the fonts is not homogeneous: for example, the font’s size of P-cofilin is smaller than the font size of cofilin => changed
Other suggestions
P2L71 “A testis‐specific LIMK2 isoform, tLIMK2, has also been described in mice [18].” The reference(18) is not exactly the good one, which is : Takahashi, H., Koshimizu, U., & Nakamura, T. (1998). A novel transcript encoding truncated LIM kinase 2 is specifically expressed in male germ cells undergoing meiosis. Biochemical and biophysical research communications, 249(1), 138–145. https://doi-org.insb.bib.cnrs.fr/10.1006/bbrc.1998.9094 => this reference has been added
P2L71 In the context of this review, it may be informative that the authors specify how tLIMK2 is different from other forms. For example: A testis‐specific LIMK2 isoform, tLIMK2, that lacks LIM domains at the N-terminus, due to usage of a testis-specific alternative initiation exon, has also been described in mice => this sentence has been modified accordingly
A transition sentence is missing L152. For instance: An activation of LIMK2 by Aurora kinase A(AURKA) has been described (ref) . As AURKA is well known => this sentence has been added
P6L217 : “This trimeric complex leads to HDAC6 inhibition and consequently to microtubule stabilization” Microtubule acetylation is not responsible of their stabilization (See Janke, C., & Montagnac, G. (2017). Causes and Consequences of Microtubule Acetylation. Current biology : CB, 27(23), R1287–R1292. https://doi-org.insb.bib.cnrs.fr/10.1016/j.cub.2017.10.044) => the text has been modified to be correct
P8L307:” LIMK localisation was also studied during mitosis [61]. This transitional sentence is inappropriate because the previous paragraph already describes the localization of LIMK1 and LIMK2 during mitosis. => the text has been modified to avoid this repetition
P9L337 “Indeed, inhibition of LIMK1 activity during mitosis leads to a delay in mitotic progression (metaphase to anaphase) and irregular spindle positioning [65,66]” : also shown in Prudent et al, 2012 => this reference was added there
P10L390-392: “Death‐associated protein kinase (DAPK), a protein implicated in programmed cell death, could act as a scaffold protein for the LIMK‐cofilin complex in TNF‐induced apoptosis, which is abrogated once cells are committed to apoptosis [72].” Not clear: what is abrogated? => we changed this sentence for a better understanding
- P15, leukaemia part : Reports of following publications are missing :
Djamai, H., Berrou, J., Dupont, M., Kaci, A., Ehlert, J. E., Weber, H., Baruchel, A., Paublant, F., Prudent, R., Gardin, C., Dombret, H., & Braun, T. (2021). Synergy of FLT3 inhibitors and the small molecule inhibitor of LIM kinase1/2 CEL_Amide in FLT3-ITD mutated Acute Myeloblastic Leukemia (AML) cells. Leukemia research, 100, 106490. https://doi-org.insb.bib.cnrs.fr/10.1016/j.leukres.2020.106490
Berrou, J., Dupont, M., Djamai, H., Adicéam, E., Parietti, V., Kaci, A., Clappier, E., Cayuela, J. M., Baruchel, A., Paublant, F., Prudent, R., Ghysdael, J., Gardin, C., Dombret, H., & Braun, T. (2022). Preclinical Evaluation of a Novel Small Molecule Inhibitor of LIM Kinases (LIMK) CEL_Amide in Philadelphia-Chromosome Positive (BCR::ABL+) Acute Lymphoblastic Leukemia (ALL). Journal of clinical medicine, 11(22), 6761. https://doi-org.insb.bib.cnrs.fr/10.3390/jcm11226761
=> these two references have been added as well as a brief description of the most striking results of these papers

Reviewer 2 Report
Villalonga E et. al. submitted an extensive review on LIMK1 and 2 kinases, their partners and regulation, substrates and physiological functions. The authors expose their involvement in cell migration, the cell cycle, the apoptosis process and neurodevelopment. In a second part, the authors review the involvement of LIMKs deregulation in several pathologies such as cancers, neuronal diseases and neurofibromatosis. The review is very dense and represent an interesting and complete summary of most literature about the subject. However, it could benefit from additional illustrations and/or tables if permitted by the journal policy. It is overall very informative and well written, with very few English typos. I therefore recommend this review for publication.
I will only give here a few examples of English typos and a misplaced sentence:
- P4, line 133: “was shown to preferentially interacts with …”
- P8, line 307: The introductory sentence “LIMK localization was studied during mitosis.” is misplaced and should be moved upward in the preceding paragraph where LIMK1 and 2 localizations in mitosis are already described.
- P10, sentence line 401-403, please add comas.
- P3: Figure 1, legend, “Nuclear export signal (NES) is shown with a black arrow,…” this sentence would be better written in plural form as there are two NES present in each LIMKs isoforms.
Author Response
We thank the reviewers for their critical and careful reading and evaluation of our manuscript. Their comments improved and enriched the submitted manuscript, which we revised accordingly. Below we provide a detailed point-by-point response to each point raised in the reviewers' reports. The line numbers refer to the revised version of the manuscript with tracked changes.
Reviewer 2
Villalonga E et. al. submitted an extensive review on LIMK1 and 2 kinases, their partners and regulation, substrates and physiological functions. The authors expose their involvement in cell migration, the cell cycle, the apoptosis process and neurodevelopment. In a second part, the authors review the involvement of LIMKs deregulation in several pathologies such as cancers, neuronal diseases and neurofibromatosis. The review is very dense and represent an interesting and complete summary of most literature about the subject. However, it could benefit from additional illustrations and/or tables if permitted by the journal policy. It is overall very informative and well written, with very few English typos. I therefore recommend this review for publication.
Suggestion of Reviewer 2 was taken into consideration as we added 5 extra figures for a better illustration and understanding of our review. We deeply thank this reviewer for this relevant recommendation.
I will only give here a few examples of English typos and a misplaced sentence:
P4, line 133: “was shown to preferentially interacts with …” => this sentence has been changed
P8, line 307: The introductory sentence “LIMK localization was studied during mitosis.” is misplaced and should be moved upward in the preceding paragraph where LIMK1 and 2 localizations in mitosis are already described. => the text has been modified to avoid this repetition
P10, sentence line 401-403, please add comas. => changed
P3: Figure 1, legend, “Nuclear export signal (NES) is shown with a black arrow,…” this sentence would be better written in plural form as there are two NES present in each LIMKs isoforms. => legend was modified

Reviewer 3 Report
Overall the review is well written and provides an acceptable up-to-date overview regarding the role of LIM kinases in different cellular events and pathologies. Bellow a suggestion and some minor corrections to improve the manuscript:
Among the LIMK physiological functions, the authors may consider mentioning the reported implication of LIMK1 in Golgi membrane dynamics/vesicle formation (PMID: 15090620, PMID: 18987335).
Lines 79, 232, and 967; italicize Drosophila => Drosophila
Line 90-91; arrow => arrowhead
Line 115; Protein kinase A (PKA)
Line 182; actin stress fiber severing => actin filament severing
Line 307; LIMK localisation was also studied during mitosis… mentioned previously (lines 299-300)
Line 454 actin depolymerizing drugs => F-actin depolymerizing toxins
LIMKi3 => LIMKi 3 (BMS-5)
Author Response
We thank the reviewers for their critical and careful reading and evaluation of our manuscript. Their comments improved and enriched the submitted manuscript, which we revised accordingly. Below we provide a detailed point-by-point response to each point raised in the reviewers' reports. The line numbers refer to the revised version of the manuscript with tracked changes.
Reviewer 3
Overall the review is well written and provides an acceptable up-to-date overview regarding the role of LIM kinases in different cellular events and pathologies. Bellow a suggestion and some minor corrections to improve the manuscript:
Among the LIMK physiological functions, the authors may consider mentioning the reported implication of LIMK1 in Golgi membrane dynamics/vesicle formation (PMID: 15090620, PMID: 18987335). => we have updated our review with these 2 references by creating a new paragraph depicting the role of LIMK1 in membrane trafficking
Lines 79, 232, and 967; italicize Drosophila => Drosophila => the text has been modified
Line 90-91; arrow => arrowhead => the legend has been changed accordingly
Line 115; Protein kinase A (PKA) => the text has been modified
Line 182; actin stress fiber severing => actin filament severing => the word has been changed repetition
Line 307; LIMK localisation was also studied during mitosis… mentioned previously (lines 299-300) => the text has been modified to avoid this repetition
Line 454 actin depolymerizing drugs => F-actin depolymerizing toxins => the sentence has been modified
LIMKi3 => LIMKi 3 (BMS-5) => the word has been modified each time it appears in the review
